

# LM4-SHARC v1.0: Resolving the Catchment-scale Soil-Hillslope Aquifer-River Continuum for the GFDL Earth System Modeling Framework

Minki Hong[1], Nathaniel Chaney[2], Sergey Malyshev[3], Enrico Zorzetto[4], Anthony Preucil[3], and Elena Shevliakova[3]

[1] Program in Atmospheric and Oceanic Sciences, Princeton University, Princeton, NJ, USA
[2] Department of Civil and Environmental Engineering, Duke University, Durham, NC, USA
[3] NOAA OAR Geophysical Fluid Dynamics Laboratory, Princeton, NJ, USA
[4] Department of Earth and Environmental Science, New Mexico Institute of Mining and Technology, NM, USA

*Correspondence to: Minki Hong (mh0093@princeton.edu)*

**Abstract.**

Catchment-scale representation of the groundwater and its interaction with other parts of the hydrologic cycle is crucial for accurately depicting the land water-energy balance in Earth system models (ESMs). Despite existing efforts to describe the groundwater in the land component of ESMs, most ESMs still need a prognostic framework for describing catchment-scale groundwater based on its emergent properties to understand its implication to the broader Earth system. To fill this gap, we developed a new parameterization scheme for resolving the groundwater and its two-way interactions with the unsaturated soil and stream at the catchment-scale. We implemented this new parameterization scheme (SHARC, or Soil-Hillslope Aquifer-River Continuum), in the Geophysical Fluid Dynamics Laboratory land model (i.e., LM4-SHARC) and evaluated its performance. By bridging the gap between hydraulic groundwater theory and ESMs' land hydrology, the new LM4-SHARC provides a path forward to learn the groundwater's emergent properties from available streamflow data (i.e., recession analysis), enhancing the representation of sub-grid variability of water-energy states induced by the groundwater. LM4-SHARC has been applied to the Providence headwater catchment at Southern Sierra, NV, and tested against in-situ observations. We found that LM4-SHARC leads to noticeable improvements in representing key hydrologic variables such as streamflow, near-surface soil moisture, and soil temperature. In addition to enhancing the representation of the water and energy balance, our analysis showed that accounting for groundwater convergence can induce a more significant hydrologic contrast with higher sensitivity of soil water storage to groundwater properties in the riparian zone. Our findings indicate the feasibility of incorporating two-way interactions among groundwater, unsaturated soil, and streams into the hydrological components of ESMs and further need to explore the implications of these interactions in the context of Earth system dynamics.

## 1. Introduction

The significance of understanding the relationship between the hydrologic cycle and climate variability has been increasingly recognized in a warming climate (Milly et al., 2008). As a major component of the terrestrial



water cycle, the groundwater (i.e., water in saturated zones beneath the land surface) plays a pivotal role in
developing the surface thermal energy and moisture dynamics and the land-atmosphere coupling by regulating how
the water and thermal energy is stored and transported across the landscape (Andreae et al., 2002; Mu et al., 2011;
Mccabe et al., 2008; Gentine et al., 2019; Fan, 2015). The effects of the groundwater state (e.g., storage) on the
water-energy balance at the land surface have been discussed in several studies based on an explicitly treated
groundwater scheme (Miguez-Macho et al., 2007; Liang et al., 2003; Yeh and Eltahir, 2005; Zeng et al., 2016;
Maxwell et al., 2011; Maxwell and Kollet, 2008). The studies commonly found higher sensitivity of surface energy
balance to groundwater storage if the water table is shallow (e.g., the water table depth less than 5 m).

However, many Earth system models (ESMs) only represent a few top meters of soil, and most, if not all,

ignore two-way interaction between the groundwater and other components of the hydrologic cycle. Specifically,
ESMs lack proper land surface scheme(s) to represent the two-way interactions between the groundwater and
stream/river while conserving the hydraulic continuity between them. In most cases, the river routing module
implemented in the ESMs exists mainly for their traditional purposes (e.g., linking precipitation-induced runoff to
the ocean) without the ability to consider the groundwater variations driven by the horizontal hydraulic gradients
between the stream and groundwater (Li et al., 2013; De Rosnay et al., 2002; Lawrence et al., 2011; Lawrence et al.,
2019; Miguez-Macho and Fan, 2012; Best et al., 2011; Takata et al., 2003).

Furthermore, the resolution of climate/water-related information provided by the current ESMs is too low

to characterize hydrological extremes and address the stakeholders' need to evaluate potential impact of future
climate change. Even an atmosphere-land water information produced at the resolution of 0.25° (Delworth et al.,
2020) — considered the most high-resolution in ESMs — is still not sufficiently fine to generate fine-scale data for
decision-making. The land components of ESMs generate the streamflow estimates only at a grid-scale (typically
ranging from 0.5° to 1.0°) (Oleson et al., 2013; Miguez-Macho et al., 2007; Fan et al., 2007; Pappenberger et al.,
2012; Campoy et al., 2013) and since such coarse-resolution streamflow data, for example, is not suitable to directly
use for locating hydrologic events at a fine-scale even in case of extremes (e.g., flooding), the scale mismatch
(between the stakeholders and ESMs) could reduce the usability of the ESM-derived projections. Another challenge
lies in the high degree of land heterogeneity with respect to soil and topographic characteristics, which needs to be
resolved at the ESMs' grid-scale. Considering the significant impacts of the fine-scale soil/topographic properties on
the hydrologic processes, the two-way interactions between the groundwater and the rest of the hydrologic cycle
must be parameterized at the sub-grid scale (e.g., catchment) to adequately represent the sub-grid variability of
hydrologic states and its implication for the broader Earth system processes and interactions (Gleeson et al., 2020;
Xu et al., 2023; Maxwell and Kollet, 2008; Pokhrel et al., 2013).

Catchments provide an appropriate scale to properly capture such sub-grid spatial heterogeneity and its

effect on the interactions among the hydrological cycle components (Clark et al., 2015; Fan et al., 2019; Blyth et al.,
2021). This is mainly due to the suitability of the catchment-scale in describing the topographically-driven flow
characteristics of water and other major fluxes (e.g., thermal energy) caused by water transport. In fact, catchments
have been considered hydrologic spatial units (or response units) where the theoretical conceptualization of surface
and subsurface water transport can be tested with readily available observational data, such as streamflow



measurements (Sivapalan, 2006; Troch et al., 2013; Kirchner, 2009). Over the past decades, many efforts have been
made to depict the hydraulically-based interactions among the distinct flow domains at the hillslope and catchment
scales (Kollet and Maxwell, 2006; Niu et al., 2011; Shen and Phanikumar, 2010; Gochis, 2021). These efforts
mainly aimed at representing the water-energy coupled balance while considering the time-dependent non-linear
relationship between water states and fluxes based on Darcian flow (Clark et al., 2015; Fan et al., 2019).

However, more specifically concerning the representation of the groundwater and its interactions with the

overlying soil and the stream/river, the spatial heterogeneity of groundwater properties (e.g., diffusivity, or effective
(drainable) porosity) remains a significant challenge for enhancing the predictability of terrestrial water storage and
exchange fluxes (Clark et al., 2009; Jing et al., 2019). Notably, an accurate description of groundwater
properties/processes at the catchment-scale is crucial because the hydrologic convergence (to the riparian zone) and
divergence (from the hilltops) by the groundwater movements significantly affect the catchments' water and energy
dynamics (Fan, 2015; Miguez-Macho and Fan, 2012; Chen and Hu, 2004; Maxwell et al., 2007). Even in existing
studies to capture the interactions among the soil-groundwater-stream, the properties of the groundwater were
treated as constant and not as emergent and time-evolving properties affected by climate or human activities (e.g.,
land use and cover changes). For example, these studies empirically fit coefficients for a storage-discharge function
(e.g., bucket model) or predefined soil properties dataset to parameterize the groundwater domains (Zeng et al.,
2016; Gochis, 2021; Newman et al., 2014; Kollet and Maxwell, 2008; Leung et al., 2011). Thus, a theoretical
approach to capture the dynamic, emergent properties of the catchment-scale groundwater has been largely absent in
those previous efforts.

The Dupuit-Forchheimer (DF) approximation allows the incorporation of the catchment-scale groundwater

into ESMs and accounts for its emergent properties/processes shaping the interactions with the unsaturated soil and
the stream (Dupuit, 1863; Forchheimer, 1986). According to the DF approximation, heterogeneity in groundwater
properties can be represented by effective parameters reflecting combined effects of the groundwater properties
variability on fluxes (e.g., baseflow), assuming that the lateral groundwater discharge is from the
homogeneous/isotropic (or systemically-defined) groundwater (Rupp and Selker, 2006; Strack, 1995; Beven and
Kirkby, 1979). The DF approximation is especially relevant as the heterogeneity of groundwater properties not
representable below model-specific resolution can be considered one of the challenges to the accuracy of modeled
data (Baroni et al., 2019; Maxwell et al., 2015; Niu et al., 2011; Bisht et al., 2017). The DF approximation is derived
by recasting the analytical solution of the Boussinesq equation (Boussinesq, 1903) into the power-law streamflow
recession model by Brutsaert and Nieber (1977). The method of streamflow recession analysis provides a theoretical
basis for estimating the effective groundwater properties on a catchment scale using (relatively readily) available
streamflow observations (Tashie et al., 2021; Hong and Mohanty, 2023a; Vannier et al., 2014; Brutsaert and Lopez,
1998; Troch et al., 1993; Xu et al., 2018).

The catchment-scale processes can be resolved at the ESMs' grid-scale via approaches to the sub-grid

model structure such as HydroBlocks (Chaney et al., 2016; Chaney et al., 2018). The hierarchical multivariate
clustering (HMC) approach, as implemented in LM4-HydroBlocks (Chaney et al., 2018), enables partitioning a
macroscale land domain grid (e.g., 0.5°, 1.0°) into sub-grid terrain units (termed 'tiles') (Milly et al., 2014; Zhao et





al., 2018; Dunne et al., 2020). Then, the catchment-scale hydrologic structure of the macroscale grid cell is
described based on the inter-tile connection enforced through the flow accumulation area derived from the digital
elevation model (DEM) (Chaney et al., 2018). This study presents a new parameterization of two-way Soil-Hillslope
Aquifer-River Continuum, namely SHARC (v1.0), tailored to the fourth generation of the Geophysical Fluid
Dynamics Laboratory (GFDL) land model version 4 (i.e., LM4) aiming to represent the catchment-scale two-way
interactions among the unsaturated soil (i.e., vadose zone), the groundwater, and stream/river in the GFDL Earth
system modeling framework. The new LM4-SHARC has been developed by extending the existing hillslope
hydrology scheme currently used in LM4-HydroBlocks (Subin et al., 2014; Chaney et al., 2018), and relying on the
catchment-scale hydrologic partitioning (of the macroscale grid cell) by HMC approach for the explicit treatment of
divergence fluxes among soil-groundwater-stream.

In sum, the LM4-SHARC (1) explicitly characterizes the catchment-scale groundwater while accounting

for its emergent properties such as groundwater diffusivity, (2) represents the two-way water and energy exchanges
in the vertical direction between the soil–groundwater and in the lateral direction between the groundwater–stream,
and (3) accounts for the groundwater-induced variations in surface water-energy budgets to enhance the ESM's
realism of land hydrology. This study primarily compares the two model configurations to evaluate the importance
of accurately describing the catchment-scale groundwater in the land component of ESM. As a proof of concept, we
apply the LM4-SHARC and LM4-HydroBlocks to a headwater catchment in the Sierra National Forest, Nevada, and
evaluate the modeled outputs against corresponding observations. Additionally, we discuss how the hydrologic
convergence to the river valley (and divergence from the hilltop) of liquid water and thermal energy due to the
groundwater flow could contribute to hydrologic contrast in a catchment based on simulations.

**2.  Methods**
**2.1.  Representation of the soil-groundwater processes in LM4-HydroBlocks**

In the current configuration of GFDL LM4-HydroBlocks, the impermeable bedrock is assumed to exist 10

m below the ground, and the sub-grid scale (e.g., reach) river dynamics were not represented. The lateral liquid
water flux between the saturated soil layers in each adjacent tile was considered groundwater flow. Hence, the
groundwater properties depend on surface/near-surface properties from which each soil column's soil properties are
derived (Subin et al., 2014; Milly et al., 2014). Each soil column in a catchment is composed of a vertical 1-D model
(i.e., soil-bedrock column) from canopy air down to an impermeable bedrock layer (Milly et al., 2014). The
processes resolved by the 1-D model include surface energy balance, vegetation dynamics, plant hydraulics,
photosynthesis, snow physics, and soil thermal and hydraulic physics (Zorzetto et al., 2023; Subin et al., 2014; Milly
et al., 2014). All the soil–bedrock columns are simulated with a 10 m soil depth at a 30-minute physical time step.
The soil–bedrock 10-m column is discretized into 20 layers. At the surface, the water flux boundary condition is
tentatively set to the difference between the sum of rainfall and snowmelt minus evaporation at each time step (i.e.,
time-dependent flow). The heat flux boundary condition at the surface is determined by the balance between the
turbulent and the radiative fluxes. At the bottom of the soil–bedrock column (i.e., 10 m below the land surface),
water flux is assumed to be zero, implying that the impermeable bedrock is always located 10 m below the land





surface. Due to this impermeable layer assumption, the water table is always identified within 10 m of the land
surface. A constant geothermal heat flux is prescribed at the bottom of the soil–bedrock columns (Milly et al., 2014).

The water fluxes between adjacent tiles are simulated on a layer-to-layer basis, defined following the soil

layer order from the land surface, according to the gradient of total pressure head between the corresponding soil
layers. The inter-tile heat and dissolved organic carbon fluxes advected by water transport are also represented.
While the tile located right next to the river reach interacts with the river, the flux from the tile to the river is a one-
way flux as the reach-scale streams' (corresponding to a catchment) hydrograph are not accounted for in LM4-
HydroBlocks. With respect to the flux exchange between the river and the soil–bedrock column (adjacent to the
reach), in fact, the river stage is set to zero (static pressure head in the reach). Consequently, the water flux is
invariably one-way, from the hillslope to the river.

**2.2.  Representation of the Soil-Hillslope Aquifer-River Interactions in LM4-SHARC**
**2.2.1. Two-way water transport and conservation in LM4-SHARC**

The LM4-SHARC solves the non-linear Boussinesq equation, which was derived from the DF

approximation to represent the lateral groundwater discharge flux from the hilltop to the river in a homogeneous and
isotropic unconfined groundwater (i.e., stream-hillslope interface) (Basha, 2013; Hong and Mohanty, 2023b;
Hornberger and Remson, 1970). According to the DF approximation, the unconfined groundwater flows laterally,
and the lateral discharge flux is proportional to the saturated groundwater thickness (Dupuit, 1863; Forchheimer,
1986). Therefore, the lateral hydraulic gradient is the only driver of groundwater lateral discharge fluxes as the
equipotential lines in the saturated zone are assumed to be vertical (i.e., hydrostatic). For saturated groundwater flow
in an unconfined groundwater overlaying an impermeable bedrock of slope $\theta$, the lateral groundwater flux is
estimated following equation (1).
$$q_l = -K_s \left[ \cos\theta \left( \frac{\partial N}{\partial x} \right) + \sin\theta \right] \tag{1}$$
where $q_l$ is the speed of lateral groundwater divergence flux (mm s$^{-1}$). $K_s$ is the saturated lateral hydraulic
conductivity (mm s$^{-1}$). Thus, the groundwater flow rate per unit width of the groundwater is given by $q_l N$, where $N$
is the thickness of the groundwater layer perpendicular to the impermeable bedrock (m). Inserting the flux equation
(1) into the mass continuity equation yields the Boussinesq groundwater equation (2).
$$f \frac{\partial N}{\partial t} = \cos\theta \frac{\partial}{\partial x} \left( K_s N \frac{\partial N}{\partial x} \right) + \sin\theta \frac{\partial}{\partial x} (K_s N) \tag{2}$$
where $f$ is the effective porosity of the groundwater (m$^3$ m$^{-3}$). As implied by the Boussinesq equation, the
groundwater properties are considered homogeneous across tiles, including $K_s, f$, and the bedrock slope $\theta$. $\frac{\partial N}{\partial x}$
denotes the groundwater hydraulic gradient between adjacent tiles. At each tile, the water table is determined by the
balance of the soil–groundwater and the lateral groundwater fluxes (Equation 3).



$$H_i^{j+1} = H_i^j + \left(\frac{r_i^j - q_{l_i}^j}{\rho f}\right) \Delta t \qquad (i = 1, \ldots, n_{HB}) \tag{3}$$
where $H_i^j$ is the hydraulic head of the water table (m) at $i$ th height band (HB) (Figure 1) at $j$ th time step. Based on
the continuity equation, the $N - H$ relationship can be established as $H = \frac{N}{\cos\theta}$. $n_{HB}$ is the total number of HBs in the
catchment. $\rho$ is $r_i^j$ is the liquid water flux between the soil–bedrock column and the water table (mm s⁻¹) at $i$ th
height band at $j$ th time step. The physical time step $\Delta t$ of the model is 30 minutes. $q_{l_i}^j$ denotes the divergence of
lateral groundwater flux at $i$ th height band at $j$ th time step and is set to the difference between the lateral divergence
from/to HBs immediately above and below. Particularly, in the HB₁ (i.e., the nearest height band to the river reach),
$q_{l_1}^j$ is determined by the balance of the groundwater discharge from HB₂ and the baseflow (if gaining reach) or the
channel infiltration (if losing reach). If multiple tiles exist in an HB, $r_i$ is effectively calculated by arithmetically
averaging the $r$ values from each tile belonging to the $i$ th height band.
LM4-SHARC also represents the reach-scale streamflow dynamics, and the resulting hydrograph is used as
the time-dependent boundary condition at the interface between the stream and hillslope. The Saint-Venant
continuity equation with kinematic wave assumption is solved for the river dynamics. For a kinematic wave, the
momentum equation assumes that the energy grade line is parallel to the streambed (Equation 4) (De St Venant,
1871; Strelkoff, 1970). Based on Manning's equation, the momentum equation can be recast in the relationship
between the steam discharge $Q$ (m³ s⁻¹) and the cross-section area of flow $U$ (m²) (Equation 5) (Manning et al.,
1890). Together with the continuity equation, the time derivative of the steam discharge ($\frac{dQ}{dt}$) with lateral influx $q_l$ (as
defined in Equation 1) can be expressed as Equation 6.
$$S_0 = S_f \tag{4}$$
$$U = \alpha Q^\beta \tag{5}$$
$$\frac{dQ}{dy} = \frac{\partial Q}{\partial y} + \frac{\partial Q}{\partial t}\frac{1}{c_k} = \frac{2q_l H_1}{\rho} \tag{6}$$
where $c_k$ is the kinematic wave celerity (i.e., $c_{k=\frac{dQ}{dU}}$ (m s⁻¹)), and $y$ denotes the river flow direction coordinate. $H_1$ is
the vertical thickness of the groundwater at HB₁, which effectively defines the wetted perimeter at the stream-
hillslope interface (m). Then, the reach outflow (i.e., discharge at the catchment outlet) was used to inversely
estimate the river stage, which was in turn used to determine the lateral hydraulic gradients between the river stage
and the water table in HB₁.
We changed the soil column's bottom boundary condition from zero-flux to a variable-flux boundary
condition so as to allow for the two-way interaction between the soil columns and the newly introduced groundwater
domain. As shown in Figure 1, due to the hydraulic gradient between the bottommost soil layer and the water table,



the vertical liquid flux ($r$ in Equation 3) defines the soil columns' bottom boundary condition if the water table is
deeper than 10 m from the land surface (Equation 7 and 8). However, if the water table is within 10 m from the
ground, the flow rate at the soil base is considered zero since the equipotential line is assumed to be vertical in the
saturated zone, so the vertical hydraulic gradient is zero (Equation 7). The chosen variable-flux boundary condition
allows the soil bottom drainage (SBD) at 10 m depth. This enables the consideration of the effects of groundwater
on the unsaturated soil processes, depending on groundwater configuration, such as groundwater properties and the
water table depth.
$$r_i^j = \begin{cases} \dfrac{K_{stog,i}^j\left(\psi_{btm,i}^j + \Delta L_i^j\right)}{\Delta L_i^j} & (water\ table\ depth > 10\ m) \\[2mm] 0 & (water\ table\ depth \le 10\ m) \end{cases} \tag{7}$$

$\Delta L_i^j = e_{btm,i} - \left(H_i^j + hl_i tan\theta\right)$     (8)
where $r_i^j$ is the vertical water flux (mm s⁻¹) and the $K_{stog,i}^j$ is the hydraulic conductivity between the unsaturated soil
bottommost layer and the water table at $i$ th height band at $j$ th time step (mm s⁻¹), which is calculated by the harmonic
mean of hydraulic conductivity values in the bottommost soil layer and groundwater. $\psi_{btm,i}^j$ is the soil matric
potential at $i$ th height band at $j$ th time step (m), $e_{btm,i}$ the elevation of the central node in the bottommost soil layer
(m). $wt_i^j$ is the pressure head of the water table at $i$ th height band at $j$ th time step (m). $hl_i$ is the total hillslope length
from the reach of $i$ th height band characterized by the HMC approach (m). $\Delta L_i^j$ is the distance between the soil
bottommost node and the water table at $i$ th height band at $j$ th time step (m).




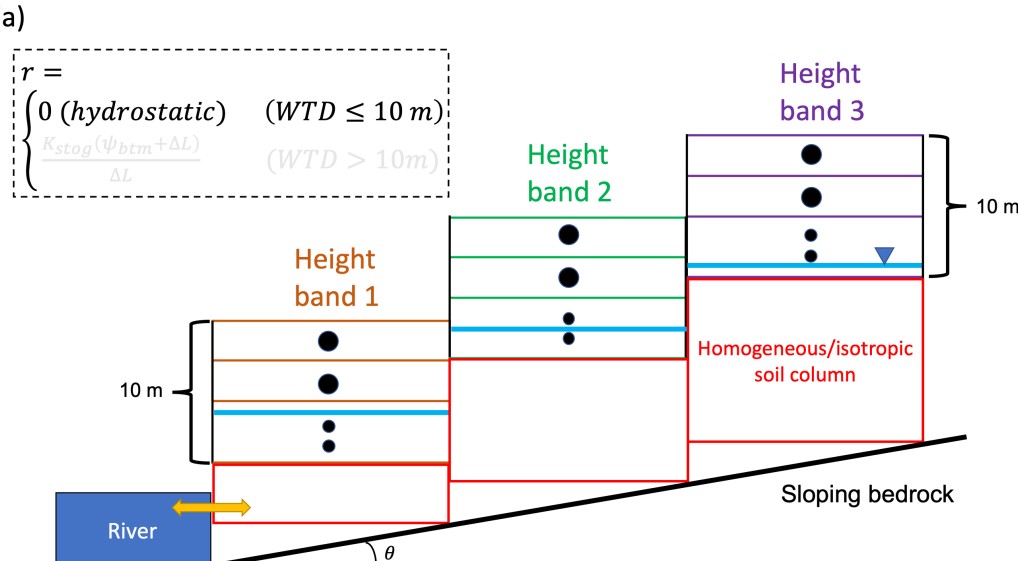

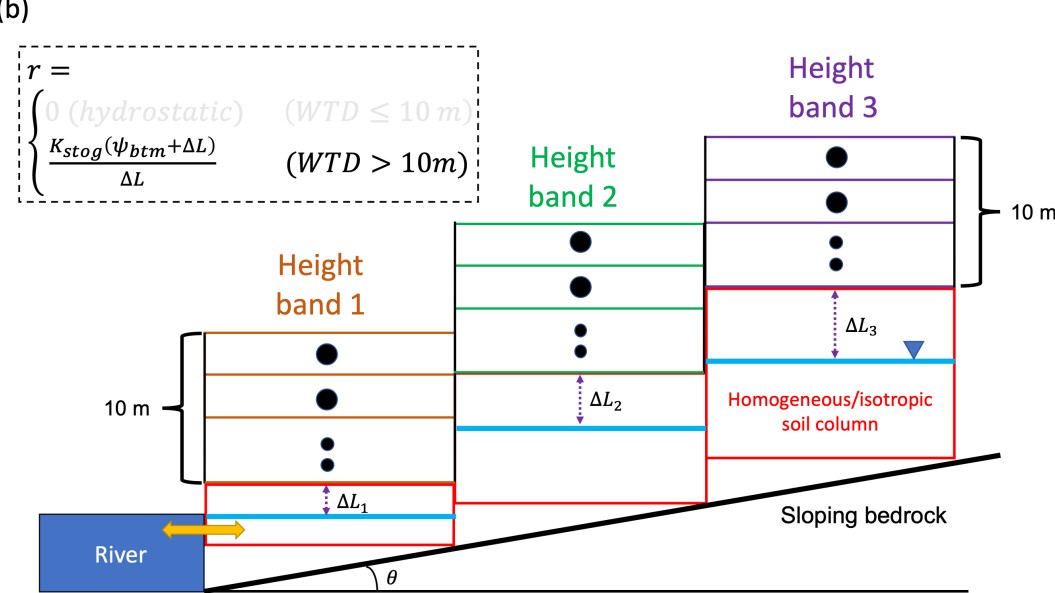

**Figure 1**. The LM4-SHARC's modified boundary condition (BC) at the soil base depending on the depth of water table. (a) the equipotential line assumed to be vertical (i.e., hydrostatic) if the depth of water table is less than 10 m, (b) the soil base BC was changed from zero-flux to variable-flux according to the hydraulic gradient between the bottommost soil layer and water table. WTD denotes water table depth.



**2.2.2. Two-way energy transport and conservation in LM4-SHARC**
LM4-SHARC accounts for the phase change of water in the groundwater domain by calculating the ice
content according to the groundwater temperature (Equation 9).
$$
ws_{gw,i}^{j+1} =
\begin{cases}
ws_{gw,i}^{j} - \min\left(ws_{gw,i}^{j}\left(T_{gw,i}^{j} - T_{freeze}\right)\dfrac{hc_{gw,i}^{j}}{hf}\right) & (ws_{gw,i}^{j} > 0\,,\ T_{gw,i}^{j} > T_{freeze}) \\[2ex]
ws_{gw,i}^{j} + \min\left(wl_{gw,i}^{j},\left(T_{freeze} - T_{gw,i}^{j}\right)\dfrac{hc_{gw,i}^{j}}{hf}\right) & (wl_{gw,i}^{j} > 0,\ T_{gw,i}^{j} < T_{freeze})
\end{cases}
$$
(9)

where $wl_{gw}$, $ws_{gw}$ are the liquid and ice content in the groundwater, respectively (-). $T_{gw}$ is the groundwater
temperature (K), and $T_{freeze}$ the freezing point of 273.15 K. $hf$ is the latent heat of fusion, a constant of $3.3358 \cdot 10^{5}$
J kg$^{-1}$. $hc_{gw}$ is the groundwater heat capacity, calculated by Equation 10.
$$hc_{gw,i}^{j} = hc_{btm,i}^{j} \times \left(e_{btm,i} - hl_i tan\theta\right) + clw \times wl_{gw,i}^{j} + csw \times ws_{gw,i}^{j}$$
(10)

where $hc_{btm,i}^{j}$ the dry soil heat capacity of the bottommost soil layer at $i$ th height band at $j$ th time step (J K$^{-1}$ m$^{-2}$).
$clw$ is the specific heat of liquid (4218.0 J kg$^{-1}$ K$^{-1}$), and $csw$ is the specific heat of ice (2106.0 J kg$^{-1}$ K$^{-1}$). We also
note that each HB-scale groundwater domain's thermal properties were assumed to be identical to those of the
corresponding soil column's bottommost soil layer.
The groundwater temperature in each HB is dynamically updated, taking into account time-dependent heat
capacity and heat fluxes conducted and advected from/to adjacent flow domains. Similar to the water table-
dependent boundary condition at the soil base, the heat flux boundary condition at the soil base is also affected by
the groundwater condition since the water advection is zero if the water table depth is less than 10 m (Equation 11,
12, and 13).
$$
\delta_{v_i}^{j} =
\begin{cases}
\left(\delta_{v_{adv,i}}^{j} + \delta_{v_{cnd,i}}^{j}\right) & (water\ table\ depth > 10\ m) \\[1ex]
\delta_{v_{cnd,i}}^{j} & (water\ table\ depth \le 10\ m)
\end{cases}
$$
(11)

$$\delta_{v_{cnd,i}}^{j} = \lambda_{stog,i}^{j}\left(T_{gw,i}^{j} - T_{btm,i}^{j}\right)$$
(12)

$$
\delta_{v_{adv,i}}^{j} =
\begin{cases}
\dfrac{r_i^{j}}{\rho}\left(T_{btm,i}^{j} - T_{freeze}\right)\dfrac{hc_{gw,i}^{j}}{\Delta t \Delta L_i^{j}} & (r > 0) \\[2ex]
\dfrac{r_i^{j}}{\rho}\left(T_{gw,i}^{j} - T_{freeze}\right)\dfrac{hc_{gw,i}^{j}}{\Delta t \Delta L_i^{j}} & (r \le 0)
\end{cases}
$$
(13)

where $\delta_{v_{cnd,i}}^{j}$ is the vertical heat conduction flux and $\delta_{v_{adv,i}}^{j}$ is the advected heat flux between the soil column and
groundwater (J m$^{-2}$ s$^{-1}$). $\lambda_{stog,i}^{j}$ is the thermal transmittance (W m$^{-2}$ K$^{-1}$) between the bottommost soil layer and



groundwater at $i$ th height band at $j$ th time step. $T_{gw,i}^j - T_{btm,i}^j$ is the temperature difference between the bottommost
layer and groundwater. The direction for $\delta_{v_{adv,i}}^j$ is determined by the water flux (i.e., downward recharge or upward
capillary) according to the hydraulic gradient. Thus, the soil columns and groundwater temperatures are simulated
with the modified heat flux boundary condition (at the soil column base) from constant thermal to variable-thermal
fluxes in the LM4-SHARC. The lateral heat transport in the groundwater domain ($\delta_{l_i}^j$) is another component in
determining the groundwater temperature (Equations 14, 15, and 16).
$$\delta_{l_i}^j = -q_{l,i}^j clw(T_{gw,i}^j - T_{freeze}) \qquad\qquad (i = n_{HB})$$    (14)
$$\delta_{l_i}^j = clw[q_{l,i+1}^j(T_{gw,i+1}^j - T_{freeze}) - q_{l,i}^j(T_{gw,i}^j - T_{freeze})] \qquad (i = 2,..,n_{HB}-1)$$    (15)
$$\delta_{l_1}^j = \begin{cases} clw[q_{l,2}^j(T_{gw,2}^j - T_{freeze}) - q_{l,1}^j(T_{gw,1}^j - T_{freeze})] & (q_l \geq 0) \\ clw[q_{l,2}^j(T_{gw,2}^j - T_{freeze}) - q_{l,1}^j(T_{stream}^j - T_{freeze})] & (q_l < 0) \end{cases}$$    (16)
where $\delta_{l_i}^j$ is the lateral groundwater heat flux (J m$^{-2}$s$^{-1}$). The groundwater temperature at $i$ th height band at $j$ th time
step ($T_{gw,i}^j$) is determined based on the time-varying heat capacity and the vertical/lateral heat fluxes from/to $i$ th
height band at $j$ th time step (Equation 17).
$$T_{gw,i}^{j+1} = T_{gw,i}^j - \frac{\left[(\delta_{v_{cnd,i}}^j - \delta_{v_{adv,i}}^j) + \delta_{l_i}^j\right]\Delta t}{hc_{gw,i}^j}$$    (17)

For the lateral heat exchange fluxes between the stream and HB$_1$ ($\delta_{l_1}^j$ in Equation 16), the stream

temperature is considered if the channel loses water to the riparian zone (i.e., $q_l < 0$). The stream temperature is
estimated considering how much heat flows into the stream from the hillslopes and out of it through the catchment
outlet (Equation 18).
$$T_{stream}^{j+1} = T_{stream}^j + \frac{\left[(\delta_{l_1}^j + \delta_{unsat_i}^j) - clw(T_{stream}^j - T_{freeze})\frac{\rho Q^j}{A^j}\right]\Delta t}{hc_{stream}^j}$$    (18)
where $\delta_{unsat_i}^j$ is the heat flux advection from the unsaturated soil to the reach by interflow (J m$^{-2}$ s$^{-1}$). $Q_i^j$ is the
stream discharge at the catchment outlet (m$^3$ s$^{-1}$), and $A^j$ is the flow area at the outlet at $i$ th height band at $j$ th time
step. $hc_{stream,i}^j$ is the heat capacity of the catchment outflow according to the streamflow hydrograph at the outlet (J
m$^{-2}$ K$^{-1}$). Consequently, in LM4-SHARC, states and fluxes for each domain are determined by accounting for the
two-way water/heat exchanges among the unsaturated soil, the hillslope aquifer, and the river.

**2.3.  The streamflow recession analysis for the groundwater parameterization**





Brutsaert and Nieber (1977) showed that the time derivative of the recession hydrograph can be expressed
as a function of streamflow $Q$ (Equation 19). Since the analytical solutions to the Boussinesq equation can be recast
in the form of a power law, the Boussinesq groundwater can be effectively characterized based on groundwater
parameters such as $K_s$, $f$, the initial saturated groundwater thickness $T_{ini}$, and the contributing area $A$ (Hong and
Mohanty, 2023a; Rupp and Selker, 2006; Brutsaert and Nieber, 1977; Szilagyi et al., 1998; Tashie et al., 2021; Hong
and Mohanty, 2023b).
$$-\frac{dQ}{dt} = aQ^b \qquad (19)$$

where $b$ is a constant, and $a$ is a function of groundwater properties. Since the geometric similarity of a unit-width
Boussinesq groundwater throughout the entire catchment is assumed, the catchment outflow $Q$ was estimated as
$Q = 2q_l H L_{reach}$ (where $L_{reach}$ is the length of the reach). This geometric similarity assumption is also reflected in
the numerical estimation of the catchment outflow hydrograph (Equation 6). Because the recession parameters $a$ and
$b$ are readily estimated by logarithmic regression of $Q$ on $-\frac{dQ}{dt}$, streamflow observations can be used to infer the
effective groundwater properties.

**2.3.1. Selecting analytical models**
Theoretical catchment outflow from the Boussinesq groundwater yields two hydraulic regimes: the early
(i.e., high flow) and late time (i.e., low flow) domains. Since the LM4-SHARC considers sloping groundwater, we
only consider the analytical solution of the Boussinesq equation for a sloping groundwater. We used the analytical
solutions obtained by Brutsaert (1994), considering their applicability for a broader range of bedrock slopes
(Pauritsch et al., 2015). Thus, the recession parameter $a$ for the early time domain is expressed in Equation 20 with
$b_{early}$ being set to 3.0, and the parameter $a$ for the late time domain is defined in Equation 21 with $b_{late}$ being set to

302 1.0.

$$a_{early} = \frac{1.33}{K_s f T_{ini}{}^3 L^2 \cos\theta}, \qquad b_{early} = 3.0 \qquad (20)$$

$$a_{late} = \frac{\pi^2 p K_s T_{ini} L^2}{f A^2} \cos\theta \left[ 1 + \left( \frac{\frac{B}{T_{ini}} \tan\theta}{\pi p} \right)^2 \right], \qquad b_{late} = 1.0 \qquad (21)$$

where $A$ is the subsurface drainage area (m$^2$) that effectively contributes to the recession slope characteristics, and $p$
is a constant set to 1/3. $B$ is the contributing groundwater's characteristic length (m) under the geometric similarity
assumption, calculated by $B=A/(2L)$.

**2.3.2. Event-scale recession analysis**
This study recognized that the recession parameters from the point cloud data (i.e., collective recession
data) could be artifacts of the variability in individual recession events (REs) (Jachens et al., 2020; Tashie et al.,



2020; Karlsen et al., 2019). To fill this gap, we performed an event-scale recession analysis to account for the
variability of recession slope characteristics among individual REs, which results in different estimates of
groundwater properties. The onset of the REs was starting 5 days after the peak to exclude the influence of overland
flow (i.e., runoff) on streamflow hydrograph. We examined the consecutive decline of daily discharge observational
data to decide the duration of an RE, and the end date of an RE was determined when the daily stream discharge was
at its lowest.

For each RE, we performed the logarithmic regression of $Q$ on $-\frac{dQ}{dt}$ in bi-logarithmic space. The time

derivative of $Q$ $(\frac{dQ}{dt})$was computed based on the daily streamflow. Each RE's transition point of the hydraulic regime
from the early to the late time domain was identified daily through the method suggested by Hong and Mohanty
(2023a). In this method, the transition point from the early time to the late time domain is determined based on an
abrupt and most noticeable change in $R^2$ values from linear regression with a fixed slope of 3.0 while incrementing
the number of the data pair of $\log(-\frac{dQ}{dt})$-$\log(Q)$ in descending order in the bi-logarithmic space. To this end, we
selectively used the daily streamflow time series of REs that lasted for more than 20 days to get enough data pairs of
$\log(Q)$-$\log(-\frac{dQ}{dt})$ for hydraulic regime distinction. For further details on the method for identifying the hydraulic
regime transition in an event-scale recession analysis, see Hong and Mohanty (2023a).

**2.4.   Characterization of the catchment-scale domain and its heterogeneity in LM4-SHARC**

A hierarchical multivariate clustering (HMC) approach (Chaney et al., 2018; Chaney et al., 2016) was used

to characterize the catchment-scale model domain and its spatial heterogeneity. The conceptual approach uses
available environmental datasets (such as soil properties, topography, meteorology, and land cover) to characterize
the sub-grid spatial heterogeneity within each grid cell of an Earth System Model (ESM) land domain (Chaney et
al., 2018). In fact, the HMC method has been previously used in constructing land domains for LM4-HydroBlocks
to account for the sub-grid spatial variability of land properties within a regular latitude-longitude climate grid cell
(e.g., at 0.5° or 1.0° resolution). The land fraction of each grid cell in the LM4-HydroBlocks domain is partitioned
into soil, glacier, and lake components. The soil component of a grid cell is composed of hillslopes clustered in a set
of $k$ characteristic hillslopes (CH). Each CH has unique attributes, such as slope, aspect, and convexity, which are
local averages of these fields obtained by high-resolution datasets. Each CH is further partitioned into $l$ units
denoted as "height bands" (HB), which are obtained by partitioning each CH based on elevation bins of size $dh$.
Finally, each HB can be further divided into $p$ clusters (i.e., tiles), and then each tile is considered representative
element volume (REV) of soil with respect to covariates incorporated for clustering (Chaney et al., 2018). The
channels were delineated using an area threshold of 100,000 m$^2$. Three parameters $k$, $dh$, and $p$ are required in the
land pre-processing code; $k$ determines the number of CH in the land model domain, $dh$ defines the height difference
between adjacent HBs, thus determining the number of HBs in a CH. The parameter $p$ sets the number of tiles in an
HB (i.e., the intra-HB variability). For further details on the HMC algorithm, see Chaney et al. (2016); Chaney et al.

(2021).





Unlike the previous configuration of the land input dataset for a regular grid cell, this study, for the first
time, extended the existing HMC framework to generate a land input dataset for a catchment domain with an
irregular boundary. This refinement has been designed to assess the GFDL land model's performance at the
catchment level while enhancing the interpretability of the land model's hydrologic outputs. The catchment's
boundary was determined by computing the geographic extent of the contributing area that drains to a specific point
(i.e., the catchment outlet). Since the catchment areas are significantly variable (roughly ranging from sub-km$^2$ to
hundreds of km$^2$), the resolution of a digital elevation model (DEM) used for clustering the terrain characteristics
should be fine enough to capture the intra-catchment heterogeneity of terrain characteristics adequately. To this end,
the USGS 3DEP 1/3 arc-second (i.e., 10-m resolution) DEM was used in this study (Usgs, 2019).

**2.5. Experimental design for model comparison**
LM4-SHARC's new parameterization for the groundwater and its interaction with the soil and river was
evaluated based on comparing the respective baseflow and near-surface soil moisture/temperature outputs from the
retrospective run of LM4-SHARC and LM4-HydroBlocks. For spin-up, we used periodically cycled GSWP3 10-
year forcing. The GSWP3 forcing from 1901-1910 was used repeatedly in each model cycle until a steady-state in
the groundwater-related model variables by replacing the initial condition for a new spin-up with the final output
state from the previous cycle. The groundwater-related variables include 1) SMC at the bottommost soil layer, 2)
baseflow, and 3) water table, and we considered the model to reach a steady state if the simulated difference
between the end of the $n^{th}$ and $(n-1)^{th}$ cycle for the respective variables satisfied our criteria simultaneously. In this
study, we set the criteria for each variable as 0.001 m$^3$ m$^{-3}$ (0.1 %) for the bottommost layer's SMC, 0.1 m d$^{-1}$ for
baseflow, and 0.001 m for water table. We note that soil-groundwater two-way fluxes ($r$ in Equation 7) was
additionally considered in LM4-SHARC for evaluating a steady-state as it is a new variable in LM4-SHARC.
Using the confirmed steady state outputs as an initial condition, both model configurations were compared
for the evaluation period from 10/1/2003 – 9/30/2014 (WY2004 – 2014, 11 years). In this study, the in-situ
precipitation and air temperature were assimilated into the GSWP3 forcing data by being directly inserted (i.e.,
direct insertion data assimilation). This was to improve the accuracy of the model outputs, given the inconsistency
between the GSWP3 forcing and the local meteorological conditions primarily due to scale difference (i.e.,
0.5°×0.5° vs. 1 km$^2$). We considered precipitation and air temperature the most significant atmospheric variables
determining the catchment's water and energy budgets, so that the in-situ precipitation and air temperature data
obtained during the evaluation period replaced the corresponding variables in the GSWP3 forcing. LM4-
HydroBlocks and the LM4-SHARC were then operated using the identical forcing input. Two statistical metrics of
Pearson's correlation coefficient $R^2$ and $RMSE$ (Root Mean Square Error) were used to evaluate the temporal
agreement of the modeled hydrologic outputs against corresponding observations and the errors, respectively.

**3. Study area and observational data**
**3.1. Study area and the HMC parameters**



The study area is the 1-km² Providence Creek P301 headwater catchment in the Sierra National Forest,
Nevada (Figure 2 a). The P301 headwater catchment is one of eight primary headwater catchments of the Kings
River Experimental Watershed (KREW) project. The U.S. Department of Agriculture's (USDA) Pacific Southwest
Research Station initiated and operated the KREW project, part of the National Science Foundation's Southern
Sierra Critical Zone Observatory (Jepsen et al., 2016; Hunsaker et al., 2012). The eight catchments are clustered into
two groups, Providence and Bull, and, in this study, the P301 headwater catchment that belongs to Providence Creek
was selected. We selected the P301 catchment because it contains only one first-order reach with no tributaries into
the reach. Since the connectivity between the catchment and reach was required to be established for developing the
LM4-SHARC model, the hydrologic configuration of the P301 catchment was considered ideal for evaluating the
developed LM4-SHARC model.
Surface elevations in the P301 catchment range from 1755 to 2114 m (Hunsaker et al., 2012), on an
average topographic slope of 19° The length of the first-order reach in the P301 catchment is 1.5 km, and the
average width of the reach was approximated at 10 m to define the channel geometry. The Providence P301
catchment represents a rain-snow mixed-conifer forest site with an annual mean precipitation of 1,315 mm/year. The
site has a Mediterranean climate, with cool, wet winters and dry summers from approximately May through October
(Safeeq and Hunsaker, 2016). During the WY 2004-2014 evaluation period, about 90 % of precipitation occurred
between November and June. The mean annual air temperature was measured at 6.8°C. Precipitation falls as a mix
of rain and snow, and precipitation transitions from majority rainfall to majority snow approximately at 2000 m in
elevation (Bales et al., 2018; Hunsaker et al., 2012).
In compiling the land input dataset for the given catchment, the $k$ was set to 1 as the study area is a single
headwater catchment. The surface elevation data from 3DEP DEM was used as the sole variable to account for the
intra-catchment variability of terrain properties, and each HB's extent was determined using a $dh$ value of 20 m. $p$
was set to 1; however, we note that the number of tiles in each HB can increase (or decrease) depending on factors
such as natural mortality, land use, and fire events applied to each tile. The stream flowline was also delineated
according to the 3DEP DEM using the area threshold of 100,000 m². Consequently, the P301 catchment was
clustered into six HBs with a delineated area of 0.9904 km² and a stream length of 1.3 km, nearly identical to the
field measurements (Figures 2 b, c, and d).



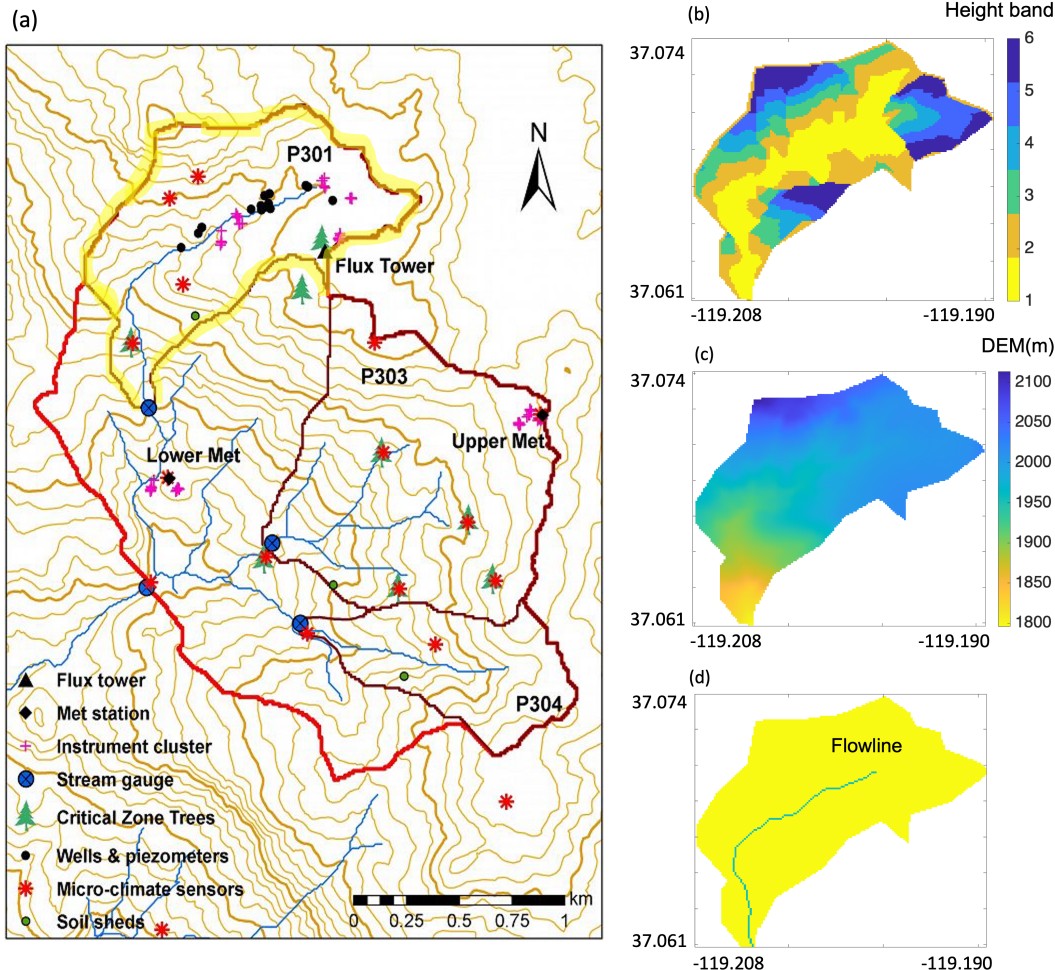


**Figure 2**. (a) The yellow highlighted area indicate the spatial extent of the headwater catchment P301. While the soil moisture
and temperature data measured at the Lower Met station is not within the P301 catchment, the measurements from the Lower
Met station was used due to its proximity to the catchment and superior quantity and continuity of observational data compared to
other measurement points within the P301 catchment, (b) By the HMC method, the P301 catchment was clustered into six
different height bands (HBs), (c) the digital elevation map (DEM) of the P301, and (d) the flowline delineated by the land input
dataset pre-processing.

### 3.2. Observations for models evaluation

### 3.2.1. Streamflow and baseflow

The streamflow observations measured at the outlet of the P301 catchment were used. The primary stream

height measurement device is an ISCO 730 air bubbler (Teledyne Isco). Backup stage measurements were initially

obtained using an AquaRod capacitance water-level sensor (Advanced Measurements and Controls, Inc.) or a Telog

pressure transducer (Trimble Water, Inc.). The stream height is measured at 15-minute intervals and converted to the





discharge rate using the standard rating curve supplied by the flume and weir manufacturers (Bales et al., 2018). The
stream discharge monitoring began in September 2003 (i.e., WY 2004). The streamflow was averaged daily from
WY 2004 to WY 2014 (i.e., 10/01/2003 – 09/30/2014, 11 years in total) and used to evaluate the daily-basis
simulation outputs in this study. The daily streamflow observational data derived the daily baseflow rate using the
baseflow separation method suggested by Szilagyi and Parlange (1998). The applied baseflow separation method
assumes that the drainage from the Boussinesq groundwater maintains the stream recession flow. Thus, the baseflow
separation method ensures consistency between the applied analytical/numerical models and the observational
baseflow data.
**3.2.2. Soil moisture, soil temperature, and snow depth**
The soil volumetric water content (i.e., soil moisture content (SMC)) and soil temperature (ST) were
measured at the Lower Met using ECHO-TM sensors (METER Group). The SMC and ST were measured at 10, 30,
60, and 90 cm below the soil surface, and the measurements were used as representative values for each soil depth
above and below each sensor (Bales et al., 2018). The measurements at the Lower Met station were used due to its
proximity to the P301 headwater catchment (~480 m to the outlet of the P301 catchment). The elevation difference
between the Lower Met station and the nearest drainage point (i.e., reach) is about 13 m. At the station, the sensor
nodes were installed in locations with different canopy coverage characteristics, such as drip edge, under canopy,
and open canopy, to account for the effect of shading (i.e., radiation interception due to canopy) on SMC and ST.
Snow depth was also measured at the same Met station, and the distance to snow (or soil if no snow cover exists)
was measured using an acoustic depth sensor located at 3 m above the soil surface (Judd Communication LLC)
(Bales et al., 2018). The sensors were installed in 2008 (i.e., WY 2009); however, due to the availability of the 10
cm depth SMC observations at the open canopy spot, the observation and simulations were compared from WY
2009 – WY 2012 (4 years). The SMC and ST observations measured at the depth nearest to the land surface (10 cm)
were used to evaluate the near-surface modeled outputs from the LM4-SHARC and LM4-HydroBlocks.
**3.2.3. Meteorological observations**
The meteorological data for model forcing were obtained from a weather station located at the Lower Met
station (elevation 1750 m), consistently with the SMC, ST, and snow observations. Precipitation was measured with
a Belfort 5-780 shielded weighing rain gauge (Belfort Instrument) located 3 m above the soil surface. The air
temperature sensor is 6 m above the soil surface (Vaisala Corporation) (Bales et al., 2018).
**4.   Results and discussion**
**4.1.  Suitability of the select analytical model**
We compared the analytical and numerical solutions of baseflow flux for different combinations of the
groundwater diffusivity ($D$) and bedrock slope ($\theta$) to test their agreements required for applying analytically derived
groundwater properties to the numerical groundwater domain. We defined the case of slow groundwater flow
(SLW) with $K_s$ of $2 \cdot 10^{-3}$ mm s$^{-1}$ and $f$ of 0.2, $K_s$ of $9 \cdot 10^{-3}$ mm s$^{-1}$ and $f$ of 0.05 for normal groundwater flow



(NRM), and $K_s$ of $2 \cdot 10^{-2}$ mm s$^{-1}$ and $f$ of 0.02 for fast groundwater flow (FST). In each diffusivity case, the
baseflow was also simulated for distinct bedrock slope conditions, such as a mild bedrock with a tan $\theta$ of 0.001
degree (MLD) and a steep bedrock with a tan $\theta$ of 0.2 degree (STP). The hydraulic conditions, except the $D$ and tan
$\theta$, for this gaining reach experiment are: (1) river stages at $j$ th time step at the discharge boundary follow a power
function using the initial river stage $h_s^j = h_s^0 \times t^{-0.01}$, representing a falling limb of the hydrograph after peak
discharge, (2) the initial head difference between initial saturated groundwater thickness ($N_{ini}$) and $h_s$ was set to the
half of $N_{ini}$ (i.e., $h_s^j|_{j=0} = N_{ini}/2$). The identical hydraulic conditions were applied to both analytical and numerical
simulations.

The temporal agreement and the total magnitude of groundwater divergence fluxes per unit width (i.e., $q_l N$)

were investigated during the recession duration of 15 days. As shown in Figure 3, $q_l N$ decreases as the hydraulic
gradient between the river stage $h_s$ and water table (at height band 1) reduces due to the discharging groundwater.
For the representativeness of the time series, the $R^2$ and $RMSE$ were estimated for the average $q_{lat}$ at each time step,
(i.e., $qN_{ave} = \frac{qN_{stp} + qN_{mld}}{2}$). The and $RMSE$ and $R^2$ were calculated at 0.00088 m$^2$ s$^{-1}$, and 0.98, respectively, in the
SLW case, 0.0015 m$^2$ s$^{-1}$ and 0.97 in the NRM case, and 0.0027 m$^2$ s$^{-1}$, 0.96 in the FST case. Although the gaps
between the analytical and numerical $qN_{ave}$ showed a bit of a drop in the agreements as the groundwater diffusivity
increased, we understand the similarly good estimation of the daily cumulative numerical $qN_{ave}$ and temporal
agreements ($R^2$ 0.96-0.98) could compensate for the gap and yield groundwater discharge estimates accurate enough
to model the streamflow recession. Specifically, care needs to be taken when the presented analytical model is used
to tune the numerical Boussinesq groundwater with extremely high $D$ and steeper bedrock (Figure 3 c). Except for
such hydraulically extreme cases which is unrealistic in real catchments (i.e., $D > 1.0$ mm s$^{-1}$), the numerical
simulation of the Boussinesq groundwater's discharge with analytically tuned parameters can be justifiable.



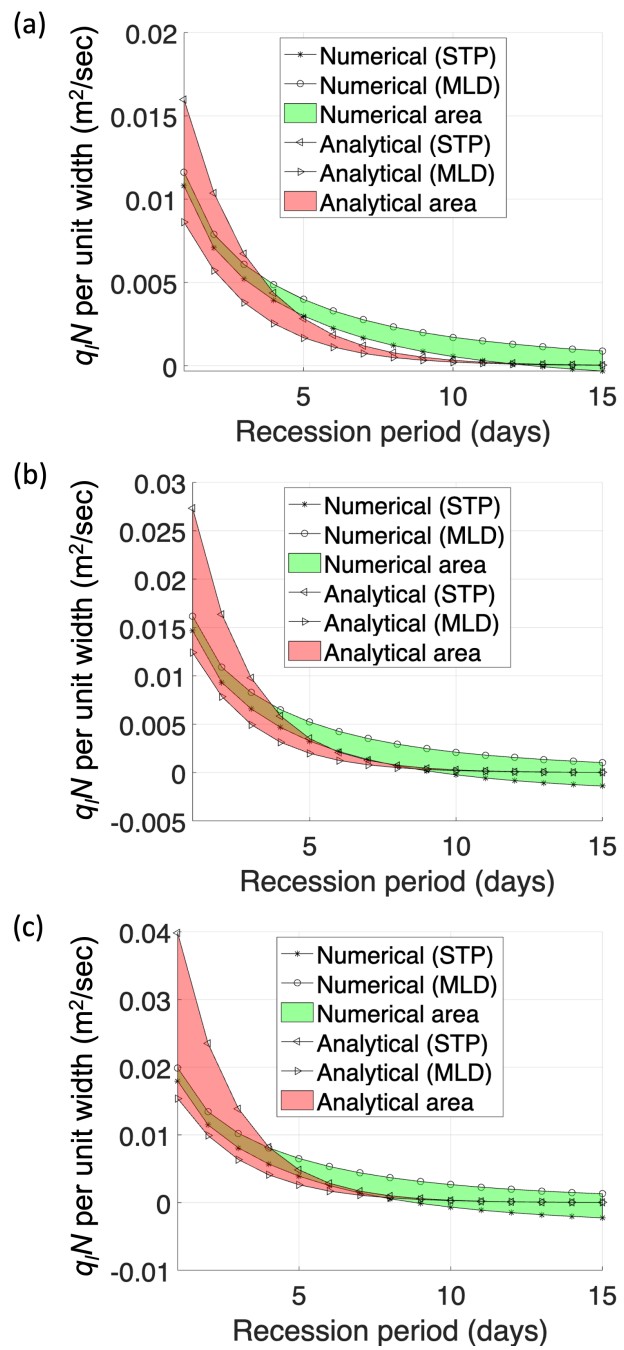

**Figure 3**. Comparison between the respective baseflow flux solutions derived from the selected analytical and numerical models.
(a) the SLW case with $K_s$ of 2.0E-03 mm/sec and $f$ of 0.20, (b) the NRM case with $K_s$ of 9.0E-03 mm/sec and $f$ of 0.05, and (c)
the FST case with $K_s$ of 2.0E-02 mm/sec and $f$ of 0.02.



### 4.2. Event-scale recession analysis and calibration

### 4.2.1. Quantifying the uncertainty of hydraulic diffusivity estimates

We performed the event-scale recession analysis to estimate the effective groundwater properties of the study catchment. Following the extraction criteria (section 3.3.2), 18 individual REs were extracted from the streamflow observations of 13 years, and their recession parameters $a$ and $b$ were estimated for each hydraulic regime (Table 1). Figure 4 (a) presents an example showing how an individual RE was analyzed with the selected analytical models for its early-time and late-time domains using daily streamflow data from the P301 catchment. Once the recession parameters $a$ and $b$ were estimated for each RE, the range of bedrock slope $\theta$ should be adequately constrained in order to determine hydraulic diffusivity parameters $K_s$ and $f$. The characteristic length $B$ is calculated by assuming the geometric similarity between both hillslope sides in the catchment; thus, $B=A/(2L)$. The initial saturated groundwater thickness $N_{ini}$ is considered constant across the individual REs, equivalent to the initial value of groundwater thickness applied in the LM4-SHARC (here, 10 m). Also, the entire catchment (i.e., 1 km$^2$) was assumed to contribute to the hydrograph recession characteristics at the catchment outlet. The following criteria were applied to define the upper and lower bound of $\theta$: (1) the effective porosity $f$ should range from 0.1% to 20.0% (Brutsaert and Lopez, 1998; Tashie et al., 2021; Troch et al., 1993; Hong and Mohanty, 2023b; Hong and Mohanty, 2023a; Heath, 2004), and (2) the catchment-scale effective groundwater lateral hydraulic conductivity $K_s$ cannot exceed 1.0 mm s$^{-1}$ (Fan and Miguez-Macho, 2011; Gómez-Hernández and Gorelick, 1989; Tashie et al., 2021).

The variability of recession characteristics was quantified by the recession parameter $a$ (since $b$ is fixed) in the early and late time domains (i.e., $a_{early}$ and $a_{late}$) (Figure 4 b). Essentially, this parameter provides insights into the variability of the groundwater's effective properties dependent on the memory effects of the catchment (e.g., groundwater storage). The 98% confidence intervals for $a_{early}$ and $a_{late}$ were estimated to be [-3.10, -2.43] and [-6.81, -6.20] respectively from the analysis of 18 REs. Following the above criteria, the value distributions of $K_s$, $f$, and $\theta$ corresponding to the ranges of $a_{early}$ and $a_{late}$ were estimated. The uncertainty of $K_s$, $f$, and $\theta$ was then quantified by determining the intersection range between $K_s$, $f$, and $\theta$ values derived from the lowest $a$ (i.e., $\log\left(-\frac{dQ}{dt}\right) = -3.10\log(dQ) - 6.81$) and the highest $a$ (i.e., $\log\left(-\frac{dQ}{dt}\right) = -2.43\log(dQ) - 6.20$). The upper and lower bounds of $K_s$ were thus estimated as [0.0026 mm s$^{-1}$, 0.0138 mm s$^{-1}$]. For $f$ the bounds were identified as [0.033, 0.190], and for $\theta$ were [5.0, 17.0 degrees]. Also, for each set of data pairs of $\log\left(-\frac{dQ}{dt}\right)$-$\log(dQ)$ with the lowest and highest $a$, the relationship between hydraulic diffusivity (per unit wetted perimeter) $D$ and the bedrock slope $\theta$ follows a power function. Consequently, we considered that the $(\theta, D)$ space in which solutions of $D$ and $\theta$ (potentially representing the catchment average behavior) could exist was further constrained by the specified range of parameters $K_s$, $f$, and $\theta$ between the power functions (Figure 4 b).





| RA | Recession period (duration day) | $\log(a_{early})$ | $\log(b_{early})$ | $\log(a_{late})$ | $\log(b_{late})$ |
|---|---|---|---|---|---|
| 1 | 4/6/04 – 5/12/04 (37 days) | -3.98 | | -6.95 | |
| 2 | 7/2/04 – 7/28/04 (27 days) | -3.72 | | -6.76 | |
| 3 | 5/20/05 – 6/8/05 (20 days) | -3.61 | | -7.10 | |
| 4 | 6/18/05 – 8/14/05 (58 days) | -3.55 | | -6.75 | |
| 5 | 5/23/06 – 6/12/06 (21 days) | -2.71 | | -6.84 | |
| 6 | 6/14/06 – 7/17/06 (34 days) | -2.99 | | -6.33 | |
| 7 | 7/30/06 – 9/6/06 (39 days) | -3.01 | | -7.66 | |
| 8 | 5/5/07 – 5/30/07 (26 days) | -2.75 | | -5.72 | |
| 9 | 6/7/07 – 7/7/07 (31 days) | -2.12 | 3.0 | -6.76 | 1.0 |
| 10 | 6/5/08 – 7/12/08 (38 days) | -1.88 | | -6.45 | |
| 11 | 7/14/08 – 8/18/08 (36 days) | -2.75 | | -5.98 | |
| 12 | 5/3/09 – 5/29/09 (27 days) | -2.12 | | -6.60 | |
| 13 | 7/1/09 – 8/5/09 (36 days) | -2.84 | | -5.61 | |
| 14 | 6/6/10 – 8/1/10 (57 days) | -1.92 | | -6.67 | |
| 15 | 8/3/10 – 8/27/10 (25 days) | -1.99 | | -6.11 | |
| 16 | 8/1/11 – 8/22/11 (22 days) | -2.86 | | -6.32 | |
| 17 | 4/27/12 – 5/24/12 (28 days) | -2.33 | | -7.12 | |
| 18 | 5/22/14 – 6/10/14 (20 days) | -2.70 | | -5.49 | |

**Table 1**. Recession period, recession characteristics, and parameters *a* and b for each recession event under different diffusivity
conditions. The variability of the parameter a indicates the variability of distinct diffusivity of groundwater across individual
recession events.




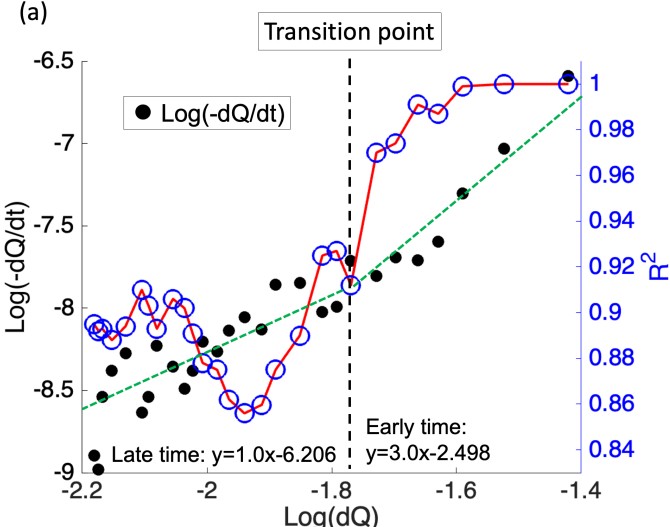

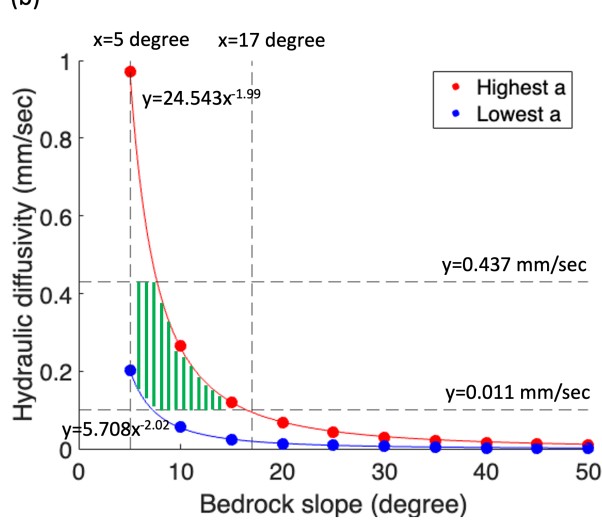


**Figure 4**. (a) An example showing how the transition point of hydraulic regime (from early-time to late-time domain) is
determined from an individual recession event. The combined understanding of the recession parameter $a_{early}$, and $a_{late}$ enables
inferring the groundwater properties such as hydraulic diffusivity $D$, (b) The uncertainty of groundwater properties $D$ and $\theta$ was
constrained by the power function relationship between them resulting from the variability of recession parameter $a$ across
individual recession events.

### 4.2.2. Calibrating groundwater properties based on baseflow flux accuracy

While it is known that groundwater properties vary across the REs due to the catchment's memory effect,
the groundwater properties that represent the long-term average behavior of the groundwater need to be tuned in the




LM4-SHARC's groundwater domain. Based on the identified value range of $\theta$ and $D,$ we tried to determine a single
pair of ($\theta$, $D$) that shows the optimal accuracy by comparing the modeled and observed baseflow data. The modeled
baseflow fluxes were estimated by summing the liquid fluxes from saturated soil to the stream, and baseflow
observations during this study's evaluation period (from 10/1/2003 – 9/30/2014, WY2004 - 2014) were used for
calibration. We considered a ($\theta$, $D$) that best represents the temporal dynamics and magnitudes of baseflow
observations as the calibrated ($\theta$, $D$) for the study catchment.

We identified that a specific parameter pair ($\theta$, $D$) can be specified in the uncertainty space. Figure 5 shows

that (1) the temporal agreements between the modeled and observed baseflow were generally related to the
magnitude predictions, (2) the most significant improvements of $R^2$ and $RMSE$ (from the LM4-HydroBlocks to
LM4-SHARC) were identified coincidentally at a specific pair of ($\theta$, $D$). We found the most pronounced
improvements of baseflow predictions in the LM4-SHARC compared to the LM4-HydroBlocks, with the $R^2$
improvement of 0.155 and $RMSE$ reduction of 0.220 mm d$^{-1}$ at $\theta$=5.0 degree, $D$=4.6· $10^{-2}$ mm s$^{-1}$. The $R^2$ and
$RMSE$ of the LM4-SHARC-derived baseflow estimates against the observations (over the 11 years) were estimated
at 0.402 and 0.556 mm d$^{-1}$, respectively, while those of LM4-derived baseflow estimates were estimated at 0.247
and 0.776 mm d$^{-1}$, respectively. With the calibrated groundwater properties, we confirmed the recession behaviors of
the P301's streamflow hydrograph over the 11-year evaluation period were generally better captured in the LM4-
SHARC's baseflow estimates compared to those of the LM4-HydroBlocks (Figure 6).



(a)

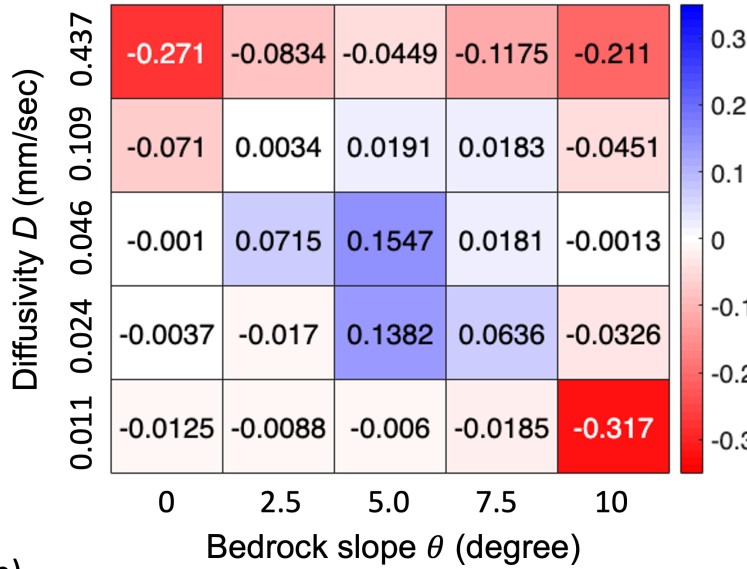

(b)

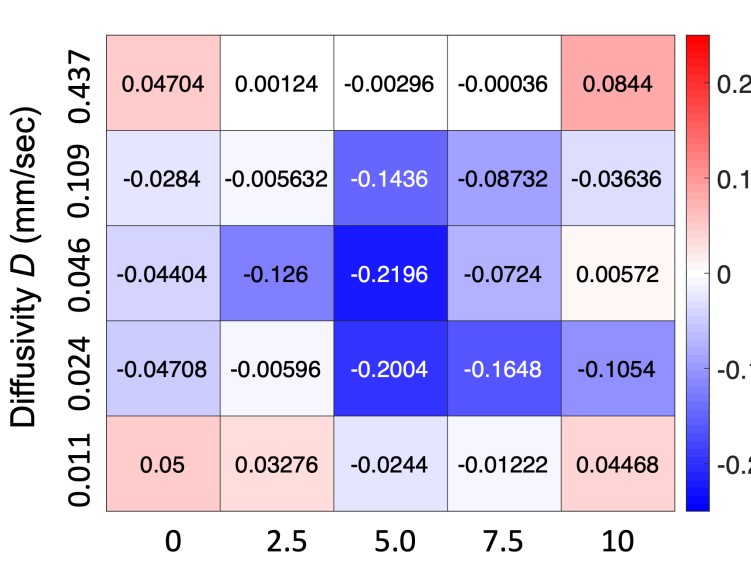

**Figure 5**. The pair of groundwater parameters ($\theta$, $D$) was specified to calibrate the catchment groundwater. (a) $R^2$ difference (improvement) between the LM4-SHARC and LM4-HydroBlocks ($R^2_{LM4\text{-}SHARC}$ - $R^2_{LM4\text{-}HydroBlocks}$). Here, $R^2$ denotes the coefficient of determination between the modeled baseflow and observations for the 11 years evaluation period, (b) for the same period, $RMSE$ difference (reduction) between the LM4-SHARC and LM4- HydroBlocks ($RMSE_{LM4\text{-}SHARC}$ - $RMSE_{LM4\text{-}HydroBlocks}$) was evaluated.



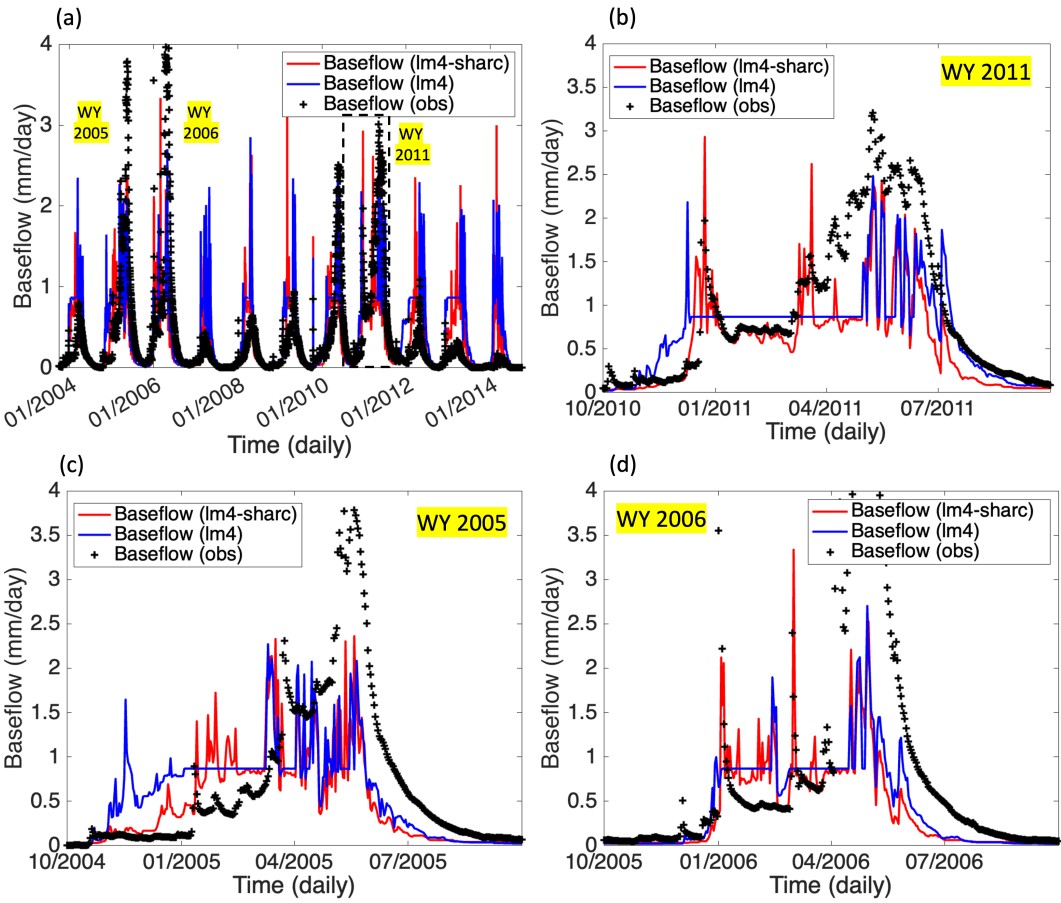

**Figure 6**. Comparative time-series of baseflow estimates from the LM4-SHARC (red), LM4-HydroBlocks (indicated LM4 in the legend, blue), and the corresponding observations. (a) Daily time-series of the baseflow data over the evaluation period of 11 years, (b) time-series in Water Year (WY) 2011, (c) time-series in Water Year (WY) 2005, and (d) time-series in Water Year (WY) 2006. The streamflow recession behavior was generally better represented in the LM4-SHARC compared to LM4-HydroBlocks.

### 4.3.  The effect of groundwater on near-surface water and energy balances

### 4.3.1. Near-surface soil moisture and temperature predictions

Based on the improved baseflow estimates, we assessed the effects of groundwater-induced soil processes on the near-surface water and energy budgets. We applied $\theta$=5.0 degree, $D$=4.6· $10^{-2}$ mm s$^{-1}$ to tune the Boussinesq groundwater domain (in the LM4-SHARC) and compared the respective soil moisture and temperature estimates at 10 cm depth from both configurations. The evaluation was performed for four years, from WY 2009-2012. Figures 7 (a), (b), (c), and (d) show the comparative time series of 10 cm depth soil moisture, soil temperature and snow mass among LM4-HydroBlocks, LM4-SHARC and in-situ observations in each WY. From all four years of the evaluation





period, we identified that soil bottom drainage (SBD) facilitation due to the modified boundary condition from zero-
flux to variable-flux BC at soil columns' base (Figure 1 a) significantly affected the soil moisture content (SMC) at
10 cm depth. The time series of 10 cm SMC from the LM4-SHARC showed significantly reduced SMC compared
to LM4-HydroBlocks. The entire soil columns reached full saturation too quickly in the LM4-HydroBlocks, and the
facilitated downward liquid transport due to drier soil lower layers in the LM4-SHARC was found to be effective in
correcting the soil's wet biases. As a result, the temporal variability and total storage of the observed SMC was
better captured in the LM4-SHARC. The average of the four $R^2$ from the yearly time series comparison (i.e., 2009-
12) was improved from 0.831 (LM4-HydroBlocks) to 0.849 (LM4-SHARC) while noticeably reducing the average
*RMSE* from 0.0722 $m^3$ $m^{-3}$ (LM4-HydroBlocks) to 0.0425 $m^3$ $m^{-3}$ (Figure 8 a and b).
We also found that the enhanced representation of SMC resulted in better capturing of near-surface soil
temperature dynamics. The decreased SMC reduces the evapotranspiration rate, especially during daytime, leading
to increased sensible heat in the soil's energy balance. Also, the reduced soil heat capacity due to the decreased
SMC (i.e., liquid water) could increase the soil temperature under the given net radiation. Consequently, we
identified that soil temperature predictions at the 10 cm depth showed significant improvements in the LM4-
SHARC, primarily during warmer seasons, and showed better agreement with the observations. The soil temperature
estimates from both model configurations showed similar values when the surface was covered by snow (i.e., snow
depth > 0). In this case the soil is insulated by snow and thus variations in soil water predicted by the two model
configurations do not lead to appreciable differences in soil temperature. From the improved skill in predicting near-
surface soil temperature in LM4-SHARC, we conclude that the soil columns with applied zero-flux BC at 10 m
depth could hold too much soil water due to the imposed shallow water table. This overestimation in soil water
content could lead to an inaccurate description of land energy balance (e.g., overestimated ET/less sensible heat),
and thus to biases in soil temperature. We note that the average of the four $R^2$ from the yearly time series
comparison between the modeled and observed soil temperature increased from 0.952 to 0.957, and the *RMSE*
significantly reduced from 2.77 K to 1.67 K (Figure 8 c and d).
The simulated snow depth from both LM4-HydroBlocks and LM4-SHARC generally showed reasonable
agreement with the measured snow depth in the catchment. We note that the melt-out timing of snow involves a
mutual influence on soil temperature. Specifically, we noticed that the timing of snow melt-out is affected by soil
temperature, as the warmer ground expedites the melting. The melt-out timings of snow in the four evaluation years
were represented sooner in the LM4-SHARC than the LM4-HydroBlocks due to increased soil temperature (Figure
7). Also, as the snow melted out, the soil temperature quickly increased as the soil was no longer insulated by snow,
leading to a higher correlation with surface air temperature.

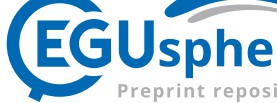




**Figure 7**. Comparative yearly time-series of soil moisture, temperature, and snow mass for the evaluation period from (a)
WY2009, (b) WY2010, (c) WY2011, and (d) WY2012. For all the years, higher 10 cm depth soil temperature values due to drier
soil were generally identified.





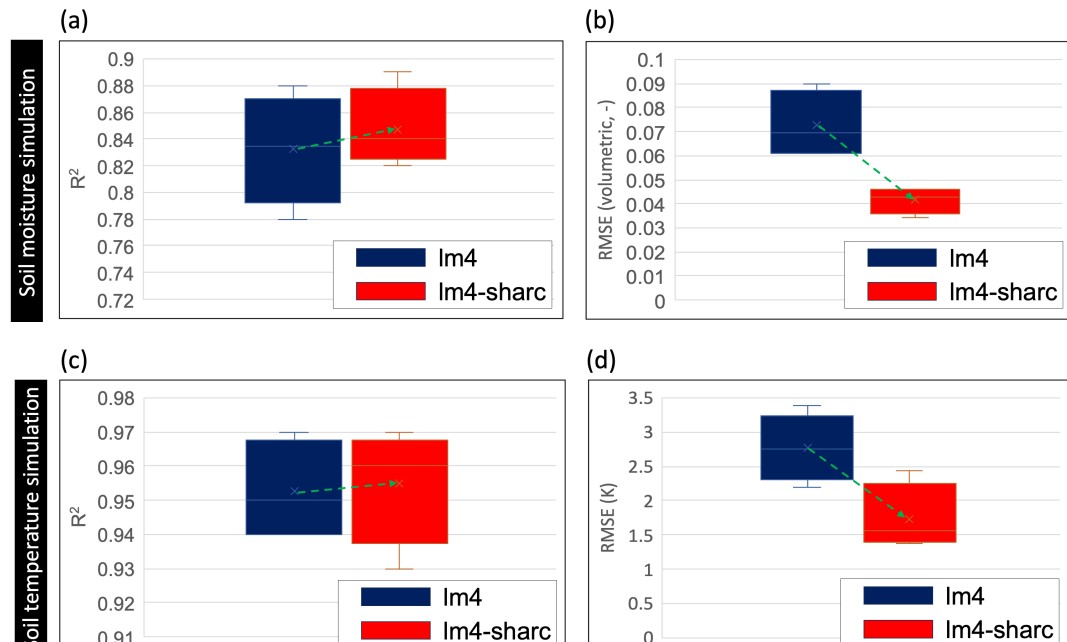

**Figure 8**. (a), (b) the average of the four $R^2$ from the yearly time-series comparison of 10 cm depth soil moisture (i.e., 2009-12) was improved from 0.831 (LM4-HydroBlocks) to 0.849 (LM4-SHARC) while significantly reducing the average *RMSE* from 0.0722 m³/m³ (LM4-HydroBlocks) to 0.0425 m³/m³, (c), (d) $R^2$ from the yearly time series comparison between the modeled and observed soil temperature at 10 cm depth increased from 0.952 to 0.957, and the *RMSE* significantly reduced from 2.77 K to 1.67 K. LM4 denotes LM4-HydroBlocks.

### 4.3.2. Sensitivity of soil water storage to stream-groundwater diffusivity

After examining the enhancement in catchment-scale water and energy balance, we further explored to what extent groundwater properties affect soil processes in LM-SHARC, focusing on soil water storage (SWS). We investigated the sensitivity of SWS to the groundwater properties $\theta$ and $D$ by estimating differences in SWS per unit area (i.e., $\Delta$SWS (kg m$^{-2}$)) between the LM4-SHARC and LM4-HydroBlocks. $\Delta$SWS was calculated by subtracting the total SWS per unit area of the soil columns (in the catchment) derived from the LM4-HydroBlocks from that of the LM4-SHARC at the end of the simulation. To this end, we need to verify that the model reaches a steady state for the given groundwater properties. The variability of SWS ($\Delta$SWS) is evaluated only after the steady state is reached (section 3.4). Figure 9 (b), (c), (d), (e) show that the simulations of the four variables reduce their deviations (from the previous cycle) as the cycles progress and how the model was considered to be steady-state based on the agreements of variables from the $n^{th}$ cycle and $(n-1)^{th}$.

We also identified that the groundwater properties influenced the time to reach a steady state (i.e., spin-up time). In addition to the three $D$ conditions considered in Section 5.1, an additional consideration of a very slow flow $D$ case (i.e., VSW, $K_s$=1.0 · 10$^{-3}$ mm s$^{-1}$, $f$=0.2) was made to investigate the expected spin-up time when the lateral groundwater discharge flux is very slow. We found that the spin-up took less time (i.e., a smaller number of





10-year repeated cycles) as $D$ increased, meaning that the LM4-SHARC reached a steady state more quickly with
the faster groundwater discharge. It turned out that the faster flux estimates could contribute more efficiently to the
convergence of other groundwater discharge-related variables (in the adjacent model domains), leading to a less
spin-up time. Likewise, from the comparison at the identical $D$ condition, the slope $\theta$ can change the spin-up time
by affecting the flux velocity. This is because the $q_l$ increases with increasing bedrock slope due to higher
quantitative growth rate of gravitationally-driven $q_l$ component compared to its diffusion-driven component. As a
result, in our estimates the spin-up times at the VSW condition range from 60 – 120 years, at the SLW condition 50
– 110 years, and at the NRM condition 40 – 80 years. We found the spin-up time in FST conditions to be 30 years,
and to be independent of $\theta$ (Figure 9 a).

ΔSWS due to groundwater discharge was investigated based on the steady state of the model. The working

hypothesis here is that the lateral groundwater discharge could facilitate the SBD by inducing the downward
hydraulic gradients from the subsurface soil and water table. Figure 10 (a) shows the variability and magnitude of
the entire catchment's annual mean of ΔSWS (over ten years) according to $\theta$ and $D$ ($y$-axis values denote average
ΔSWS per soil column). It is noticeable that the annual mean of ΔSWS gradually increased negatively (i.e., less
SWS in the LM4-SHARC) with increasing $D$ until the water table dynamics did not significantly affect hydraulic
gradients between subsurface soil-water table (i.e., lack of groundwater storage). This happened when the downward
groundwater recharge fluxes were continuously less than $q_l$ in the groundwater domain. Also, the catchment's total
ΔSWS was found to be lower if the slope is steep. While the steeper bedrock showed a greater $q_l$ with the increased
gravitationally-driven discharge fluxes, ΔSWS values were more noticeably affected by the lowered water table due
to milder bedrock in the groundwater domain. In the case of this catchment, it was also observed that for $D$ values
greater than 0.1 mm s$^{-1}$, the lack of groundwater storage occurred irrespective of the bedrock slope.

Moreover, the SBD facilitation due to the groundwater lateral discharge was more noticeable as the height

band was farther from the reach. The annual mean of ΔSWS per unit area gradually decreased from HB$_6$ (~120 kg
m$^{-2}$ y$^{-1}$) to HB$_1$ (~10 kg m$^{-2}$ y$^{-1}$), with a sharp decrease at HB1 (Figure 10 b). It was also noticeable that the distinct $D$
made less difference among the values of ΔSWS per unit area as the height band was farther from the reach. Figure
10 (c) and (d) illustrate that the effects of groundwater flow conditions on the SWS variability could be more
significant in the riparian/river valley compared to the hilltop area. This also implies that the groundwater's effects
on the water content in the partially-saturated soil depend on the depth of water table, leading to higher sensitivity of
SMC to groundwater diffusivity if the water table is shallow than deep groundwater. While the HB$_6$'s annual ΔSWS
per unit area (m$^2$) values were the greatest among the HB$_1$ – HB$_6$, ranging from 121.45 kg m$^{-2}$ y$^{-1}$ (VSW) to 121.89
mm (FST), the ΔSWS difference (between VSW and FST) of 0.44 kg m$^{-2}$ y$^{-1}$ was minor compared to the result from
the HB1. From HB$_1$, the ΔSWS values were found to range from 7.75 kg m$^{-2}$ y$^{-1}$ (VSW) to 11.70 kg m$^{-2}$ y$^{-1}$ (FST),
yielding a difference of 3.94 kg m$^{-2}$ y$^{-1}$ due to the groundwater diffusivity.


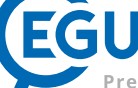

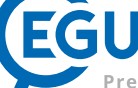

**Figure 9**. (a) varied spin-up time due to distinct groundwater properties $\theta$, and $D$. The simulations of (b) the volumetric water content at the soil bottommost layer, (c) the groundwater table, (d) groundwater recharge fluxes (since positive), (e) baseflow flux reduce their deviations (from the previous cycle) as the cycles progress to be considered steady-state based on the differences from the $n^{th}$ cycle and $(n-1)^{th}$.

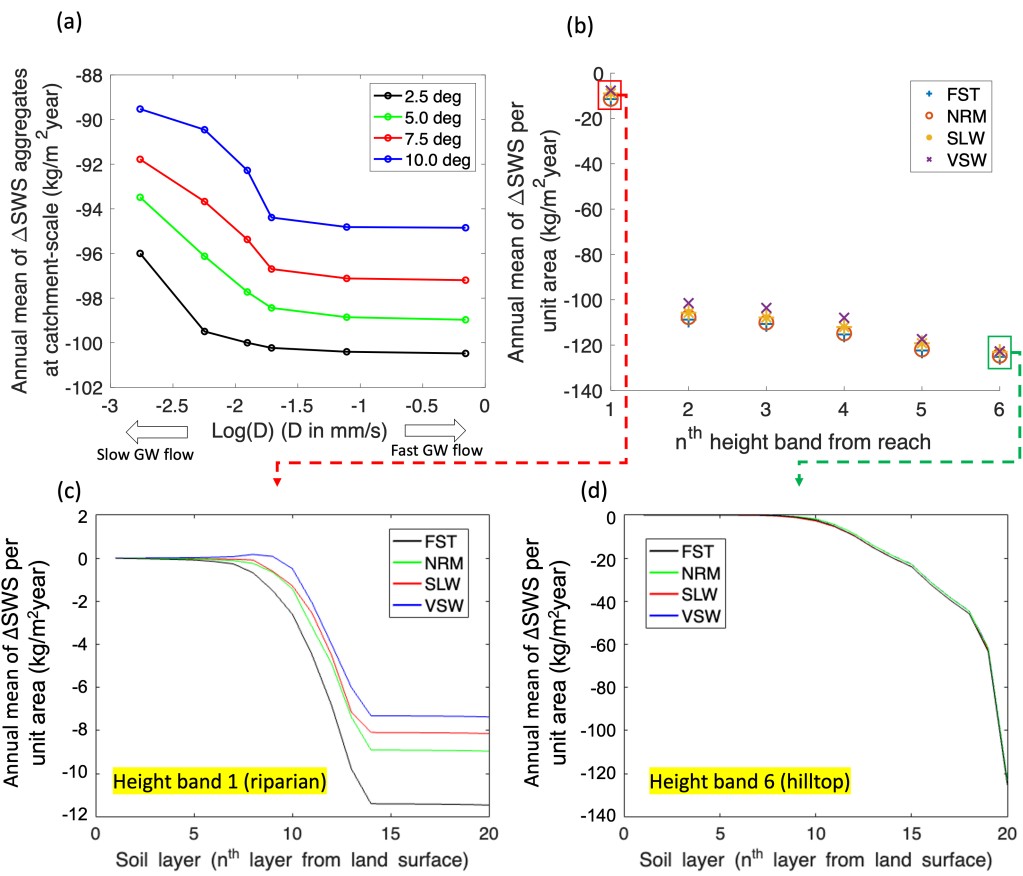

**Figure 10**. (a) Sensitivity of soil water storage (SWS) to the diffusivity $D$ and bedrock slope $\theta$, (b) the soil drainage (i.e., downward recharge) facilitation due to the groundwater lateral discharge was more noticeable as the height band was farther from the reach, (c), (d) a more significant effects of groundwater flow conditions on the SWS in the riparian zone compared to the hilltop area.

### 4.4. Distinct vegetation characteristics in the catchment due to hydrologic contrast

The water convergence due to the groundwater lateral flow induces the hydrologic contrast at the catchment-scale, leading to distinct vegetation characteristics depending on the distance from the river (Fan et al., 2019). Here, we used the modeled LAI to infer distinct vegetation characteristics (i.e., plant density) in the study catchment. The LAI was simulated from 1901 to 2014 (114 years) in both model configurations without spin-up using the GSWP3 forcing with assimilated in-situ precipitation/air temperature data (i.e., GSWP3 (01/1901-09/2003) + in-situ (10/2003-09/2014)). The comparison of LAI time series between the LM4-SHARC and LM4-HydroBlocks revealed that the differences in the LAI at the hilltop area (i.e., $HB_6$) were more dramatic than those at the riparian zone (i.e., $HB_1$) as the vegetation evolved (Figure 11 a and b). With a closer look at the LAI time series for the recent four WYs (i.e., WY 2009-2012 as studied in Section 5.3.1) at the riparian zone and hilltop, moreover,



we found that the LAI contrast was more significant during the warmer season at the hilltop area, while the overall
trend of the LM4-SHARC-derived LAI evolved comparably to that of the LM4-derived LAI at the riparian zone
(Figure 11 c, and d). The hydrologic convergence at the riparian zone explains the comparable LAI dynamics at the
riparian zone as the subsoil domain readily saturated by the converging water impedes SBD, leading to higher water
retention in the partially saturated soil. Different soil moisture availability (between $HB_1$ and $HB_6$) resulting from
the contrasting SBD dynamics is thus emphasized by the density of plants, especially during the warm(er) season
when the plants yield higher transpiration rates. The varied transpiration rates in the LM4-SHARC also consistently
explain the LAI contrast due to different soil moisture conditions. We found that the transpiration rate at the hilltop
was reduced by 29.5% in the LM4-SHARC, while the rate was reduced by 10.3% in the riparian zone in the LM4-
SHARC (Figure 11 e and f). Overall, we found that the variations in SWS, transpiration, and LAI simulations are
consistent in that the groundwater convergence to the river valley intensified the catchment's contrasting hydrologic
states.





**Figure 11**. Temporal evolution of LAI from 1901 to 2014 (114 years) using GSWP3 forcing at (a) the riparian zone (HB1) and (b) the hilltop (HB6). The differences in the LAI (between the LM4-SHARC and LM4-HydroBlocks) at the hilltop area were more dramatic than those at the riparian zone. (c), (d) the LAI time series for the recent four WYs (i.e., WY 2009-2012), (e), (f) the time series of plan transpiration rate (mm/day) for the same period. LM4 denotes LM4-HydroBlocks.





**4.5. Applicability of the LM4-SHARC in an ESM**
**4.5.1. Testing LM4-SHARC in various climatic and orographic zones**

To support the implementation of LM4-SHARC in the GFDL ESM, we need to investigate the

performance of LM4-SHARC in various climatic and orographic zones. For this purpose, three additional headwater
catchments were selected based on their precipitation and topographic slope characteristics (Figure 12 a): the
Musselshell (MT), Maine (ME), and Clearwater (WA) headwater catchments. The precipitation and slope
characteristics of these sites vary from those of the Clearwater catchment (the wettest: 3,136 mm y$^{-1}$, and steepest:
0.547 m m$^{-1}$ catchment) to those of the Musselshell catchment (the driest: 397 mm y$^{-1}$, and mildest: 0.094 m m$^{-1}$). In
this experiment, the groundwater properties in the additional catchments were assumed to be identical to those of the
P301 headwater catchment so that diffusivity $D$ was set to 0.046 mm s$^{-1}$ and the slope of groundwater bedrock $\theta$ was
assumed to be 5 degrees in all the three headwater catchments. The steady state of the groundwater and any adjacent
flow domains was also ensured, and the evaluation was performed using the 10-year model outputs.

We tried to identify if the model is robust under diverse conditions by examining the consistency between

hydrologic characteristics and indices. The hydrologic indices include: (1) runoff ratio (i.e., ratio of streamflow to
precipitation), (2) baseflow coefficient (i.e., ratio of baseflow to precipitation). The annual mean
baseflow/streamflow of each catchment was estimated at 10.9 mm y$^{-1}$/49.8 mm y$^{-1}$, 105 mm y$^{-1}$/364.0 mm y$^{-1}$, and
281 mm y$^{-1}$/1,897.4 mm y$^{-1}$ from the Musselshell, Maine, and Clearwater catchment, respectively. With the
respective annual mean precipitation of each catchment (i.e., Musselshell - 397.2 mm y$^{-1}$, Maine - 1223.1 mm y$^{-1}$,
and Clearwater - 3136.6 mm y$^{-1}$), each catchment's baseflow coefficient/runoff ratio was estimated at 0.028/0.125,
0.086/0.298, and 0.230/0.604 the Musselshell, Maine, and Clearwater catchment, respectively. The established
positive correlation between the baseflow coefficient and runoff ratio is consistent with what is reported in existing
studies (Cheng et al., 2021; Ouyang et al., 2018) (Table 2). We also found that the value gradients of
baseflow/recharges estimates correspond to what is expected from the slope/precipitation gradients. For example,
the highest yield of baseflow from the Clearwater catchment (i.e., 0.77 mm d$^{-1}$) can be explained by its high
precipitation and steep slope, which contribute to higher drought flow (i.e., baseflow during dry seasons) and peaks.
Also, the partially saturated soil found in most parts of the Musselshell catchment explains the minimal baseflow
amounts due to the lack of groundwater storage from the limited groundwater recharge (Figure 12 b and c).










| | | Headwater catchment | | |
| --- | --- | --- | --- | --- |
| | | Musselshell (MT) | Maine (ME) | Clearwater (WA) |
| Hydrologic characteristics | Annual mean precipitation | 397.2 mm y$^{-1}$ | 1,223.1 mm y$^{-1}$ | 3,136.6 mm y$^{-1}$ |
| | Annual mean streamflow | 49.8 mm y$^{-1}$ | 364 mm y$^{-1}$ | 1,894.7 mm y$^{-1}$ |
| | Annual mean baseflow | 10.9 mm y$^{-1}$ | 105 mm y$^{-1}$ | 281 mm y$^{-1}$ |
| | Annual mean groundwater recharge | 0.21 mm y$^{-1}$ | 150.4 mm y$^{-1}$ | 360.2 mm y$^{-1}$ |
| Indices | Runoff ratio | 0.125 | 0.298 | 0.604 |
| | Baseflow coefficient | 0.028 | 0.086 | 0.230 |

**Table 2**. The hydrologic characteristics and indices of the three additional headwater catchments from the simulated outputs with the GSWP3 forcing from 1901-1910. The annual mean values of each catchment are estimated from 10-year cycle based on confirmed steady state.



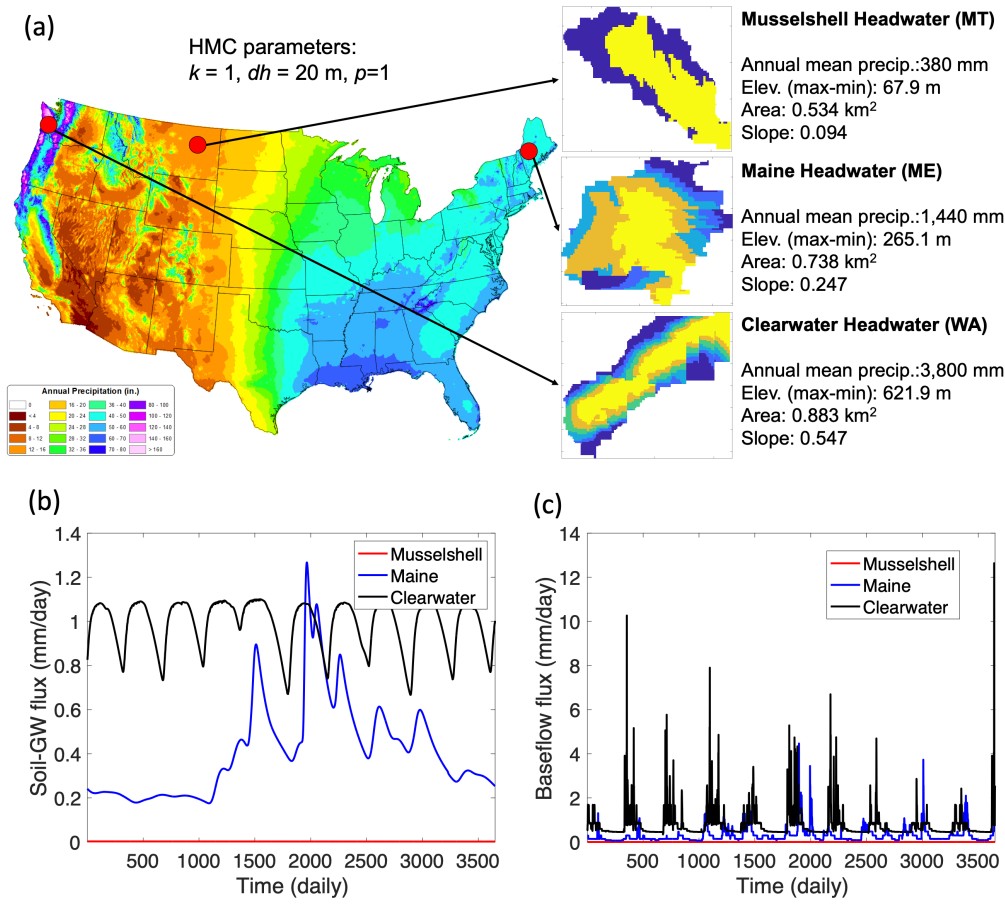


**Figure 12**. (a) the additional three sites are marked (red dots) on the PRISM 30-year normal precipitation map (2022, PRISM
Group, Oregon State University). The sites include Musselshell headwater (MT), Maine headwater (ME), and Clearwater
headwater (WA) catchments, (b) the LM4-SHARC's daily soil-groundwater exchange fluxes (mm/day) over 10 years from the
three catchments, (c) the simulated daily baseflow fluxes (mm/day) for the corresponding period and sites.


**4.5.2. Inferring groundwater properties for global-scale SHARC simulations**

Since the presented parameterization scheme SHARC relies on the observationally-derived recession

characteristics of streamflow, a method must be developed for quantifying the recession variability (using
parameters $a$) of catchments with no streamflow information. This necessity is particularly emphasized as the
SHARC scheme will ultimately be used for global simulations. While not providing specific research findings, this
section aims to discuss several possible approaches based on existing studies. Essentially, we expect that existing
global-scale datasets of soil, topography, and climatology and remotely-sensed hydrologic data can be used
complementarily to infer the ungauged catchments' recession characteristics. For example, the significant
correlation between the recession parameter $a$ and catchments' soil/geology attributes (from a global database) was



established by means of simple regression analysis or machine learning (ML) techniques (e.g., random forest)
(Zecharias and Brutsaert, 1988; Hong and Mohanty, 2023b; Tashie et al., 2021; Cai et al., 2021). ML procedure can
leverage remotely sensed hydrologic data, such as surface soil moisture (e.g., SMAP), evapotranspiration (e.g.,
MODIS), and groundwater storage (e.g., GRACE), to enhance the robustness of groundwater parameter estimation
with identified relative importance of each input variable. Moreover, the recent launch of SWOT (Surface Water and
Ocean Topography) spaceborne mission offers the potential to gather river discharge and baseflow data at a
temporal resolution of interest (e.g., daily) with unprecedented global coverage (Baratelli et al., 2018; Li et al., 2020;
Wongchuig-Correa et al., 2020). These studies aimed to overcome the spatial and temporal gaps in SWOT
observations by integrating a large-scale hydrologic model with synthetic SWOT data through techniques like data
assimilation. Based on existing findings about the utility of SWOT data for the baseflow estimation, the possibility
of obtaining baseflow and streamflow at a higher spatial and temporal resolution will be further explored to use
model-derived outputs as surrogate data to investigate the catchment-scale recession behavior with global coverage.

## 5.   Conclusions

The presented new framework LM4-SHARC harnesses the parametric efficiency of the DF approximation-

based Boussinesq groundwater in capturing the emergent properties of the catchment-scale groundwater. This study
proposed a calibration method for the catchment-scale groundwater based on the accuracy of baseflow fluxes and
also demonstrated the contribution of the additional groundwater domain/processes (with tuned groundwater
parameters) to the improvements of the catchment-scale water/energy budgets. The streamflow recession analysis
provides a physically explicit and viable way to use readily available streamflow measurements to infer the time-
evolving groundwater properties. Thus, we argued how the time-evolving groundwater diffusivity could be
considered in Earth system modeling through the combined use of numerical/explicit groundwater domain and the
observationally-derived stream discharge information.

The notable improvement in soil moisture and temperature predictions resulting from the LM4-SHARC's

hydraulic continuum scheme underscores the necessity of resolving the catchment-scale groundwater dynamics and
its interactions with the soil and the stream at the grid-scale of ESM. Specifically, our analysis shows that vertical
soil drainage to relatively deep groundwater should be taken into account when simulating soil moisture and
temperature. We also note that the soil columns in ESMs might hold too much water without the groundwater-
induced drainage dynamics. The significant amounts of facilitated drainage (i.e., $\Delta$SWS) roughly around 90-110 mm
$y^{-1}$, which corresponds to about 6-7 % of the total precipitation in our study site, also emphasizes the importance of
considering the lateral groundwater divergence (from hilltop) and convergence (to riparian zone) in ESM land
components. The existing biases in soil moisture and surface temperature can lead to a flawed description of other
land variables, impacting the surface energy balance, carbon cycle, and biogeochemistry. As the simulated soil
temperature was found to be lower than observed values due to the ratio of sensible heat to latent heat and soil heat
capacity as a function of SMC, liquid water contained in the partially saturated soil needs to be better quantified as it
significantly influences the dynamics of land surface energy balance.



Scaling the fine-scale surface water-energy processes to the GCMs' grid cell scales while properly
considering the hydrologic interactions and heterogeneity is one of the primary objectives in the ESM community.
Considering that the streamflow is a major water flux (that significantly affects energy, carbon, and biogeochemical
fluxes) crossing the catchments, in order to properly scale the effects of the SHARC's catchment-scale hydraulic
continuum to the macroscale grid cell, a reach-to-reach connection throughout the river network (i.e., stream/river
routing) needs to be established. Also, based on the enhanced baseflow production in LM4-SHARC, we expect
significant qualitative enhancements of streamflow estimates, which will, in turn, lead to enhanced surface/near-
surface water and energy budgets as well as flooding representation (e.g., floodplain dynamics). Overall, the
improved water and energy balance in LM4-SHARC is expected to be relevant for coupled land-atmosphere
simulations, where refining the land surface state plays a significant role in developing the lower atmospheric
boundary layer, and also to contribute to the efforts to address societal challenges by hydrologic extreme events such
as flooding with enhanced streamflow production.

*Author contributions*
All authors contributed to research design and manuscript editing. M.H. developed software, performed the
model evaluation, data analysis and drafted the first version of this manuscript. N.C., S.M., and E.S. contributed to
software development. E.Z., and A.P. contributed to data preparation.

*Competing interests*
A co-author is a member of the editorial board of Geoscientific Model Development (GMD).

*Acknowledgement*
This report was prepared by Minki Hong under award NA23OAR4050431I from the National Oceanic and
Atmospheric Administration, U.S. Department of Commerce. The statements, findings, conclusions, and
recommendations are those of the author(s) and do not necessarily reflect the views of the National Oceanic and
Atmospheric Administration, or the U.S. Department of Commerce. The authors thank Dr. John Dunne and Dr.
Kirsten Findell at NOAA GFDL for reviewing a first draft of the manuscript.

*Code and data availability*
The source code of LM4-SHARC v1.0 and the model input data such as model domain dataset and forcing
data are shared in a public repository: https://zenodo.org/records/13750071.

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
