# Peer review of "LM4-SHARC v1.0: Resolving the Catchment-scale Soil-Hillslope Aquifer-River Continuum for the GFDL Earth System Modeling Framework"

_EGUsphere, 2024_

## Author Comment (AC1)

**Note)** Line numbers correspond to the non-tracked changes in the copy of the manuscript.

**Reviewer Comments 2 (RC2)**

This paper presents a new parameterization scheme that cooperate the groundwater component to the ESM model, which is a good starting point of considering the effect of groundwater on the energy component in the ESM model.

In general, the manuscript is written in very good and comprehensive english and it is easy to read and follow.

However, I find the manuscript is too long and with some redundant information that reduces the readability. I do agree with the first reviewer that the part describing the energy transport makes the manuscript discontinued, which might be better to put to as a supplementary.

- We appreciate the reviewer's insightful comments. Based on the reviewer's suggestion, we have updated the section 2.3.2 to improve the readability. Specifically, explicit equations for soil, groundwater, and river temperature estimation and heat flux were retained in the main text. In contrast, all other equations (i.e., dynamic heat capacity and details of heat flux estimation) were categorized as Appendix A and excluded from the main text. As a result, section 2.3.2 has been shortened significantly while addressing the essential information of temperature simulation.

I believe with some more efforts for revision, this manuscript could reach the standard of a publication.

- We appreciate the reviewer's encouraging comments. All the specific comments made by the reviewer have been addressed in the following section.

Here follows some specific comments.

* Figure number has been changed due to the added Figure 2 – workflow diagram.

1. Figure 1 appears to be difficult to understand. especially when it appears, there was no proper explanation of what is a "height band" until I read section 2.4. I suggest to have a good explain of the following terms at the beginning: what is soil columns, tiles, height band and characteristic hillslope, respectively and what are their potential relationship. and also the meaning of the black dots and what does the size of the dots mean in each soil layer is not clear.

- We appreciate the reviewer's insightful comments. Referring to the reviewer's comments, we have moved section 2.4 to the very front of section 2, agreeing that the readers need to understand the concept and relationship among CH, HB, and tiles before those spatial units are used to explain the processes. Thus, the definitions and relationships among CH, HB, and tiles are discussed in section 2.1 in the revised manuscript. Also, Figure 1 has been updated to match the dot size. Initially, the different sizes of dots in the top two layers and others implied that there are multiple soil layers (i.e., 20 layers in total) below the top two soil layers. To avoid confusion, the dots' sizes are matched, and what the dots mean is also noted in Figure 1's caption.

2. line 183 states rho is $r\_j ^j$ , which does not make sense.  and if rho is density, then could you check the unit consisensy of this equation?

- We apologize for the typo and appreciate for pointing this out. $\rho$ is the liquid water density (1,000kg/m$^3$) and $r_i^j$ is the liquid water flux (mm/sec). We have made the correction in the revised manuscript accordingly (lines 213-214).

3. Line 196 eq 6 seems to be the spatial derivative of the "stream" discharge instead of the time derivative of the "steam" discharge? 195 has typo "steam discharge" as well.

- We appreciate the reviewer's comments. $\frac{dQ}{dy}$ in Equation 6 is the spatial derivative, not the time derivative. The typos in Equation 6 and 'steam discharge' have been fixed in the revised manuscripts (lines 237-238). We apologize for the typo.

4. Figure 3: the analytical area is not explained anywhere (or I missed it), if you present it there, it is better to give some explanations.

- The numerical and analytical area (Figure 4) means the uncertainty band of analytical solutions due to the slope variations implemented in each experiment. Figure 4's caption has been updated to clarify the meaning of the numerical/analytical area. We apologize for the missing information.

5. Line492 "Following the extraction criteria (section 3.3.2)....  there is section 3.3.2

- We apologize for the typo. We have corrected the typo in the revised manuscript (line 509). Section 3.3.2 → Section 2.4.2

6. Figure 7 (a)(b)(c): the first subfigure with legend "VWC", which is not explained anywhere. according to the caption it is the soil moisture, then should it SMC? (according to line 581). And what is the meaning of fprec in the third figure? how to interpret it?

- Thank you for pointing this out. According to the reviewer's comments, the term VWC has been replaced with SMC (m$^3$m$^{-3}$), and the meaning of fprec has also been clarified in Figure 8's caption.

7. Line 636-648 + Figure 9 : It is not clear why reducing spin-up time matters. it is an interesting discovery, but I don't find it important.  Do you have any important reason to keep this discussion in the main body of the manuscript? Otherwise consider move it as a supplementary material.

- As suggested by the reviewer, we have moved Figure 10 (a) and the corresponding sentences to Supplemental Information S.2. As a result, section 4.3.2 has been significantly shortened.

8. Section 4.4 is very important since it finally hits the point that the modeled LAI has difference after involving the groundwater to the model, which lead to the potential improvement of the modeled transpiration. It would be better if there are more statistics given to show how much improvement between the lm4 and lm4-sharc.

- We have addressed the reviewer's comments by performing an additional comparison between simulated and observed evapotranspiration. The Critical Zone Observatory (CZO) (at the Providence Creek headwater catchment) vapor flux observation from the eddy flux tower is located within the study catchment, as indicated in Figure 3. Since observations that only measure 'transpiration' are not available and the vapor flux observations measure ET (i.e., evaporation +

transpiration), we compared simulated ET against observed ET when transpiration is above zero (see Figure 12(f)). In the revised manuscript, we also note that 33.5 percent of the data during the observation period was missing, and the loss of approximately one-third of the data reduces the reliability of this comparison. All this information has been organized as Supplemental Information S.3.

9. I see all the catchments the authors selected are headwater catchment. So, before discussing applying the sharc scheme to global scale, could the author elaborate more on how to apply the scheme to downstream catchments where the drainage networks are more complicated? I think the accuracy of the analytical solution of those catchments should be more challenging.

- To address the reviewer's comments, we have revised section 4.5.2 to present the applicability of observationally-derived model data of net subsurface discharge (NSD) to parameterizing the Boussinesq aquifer's diffusivity downstream of the river (lines 799-814) (Hong and Mohanty, 2023a). Essentially, as we respond to reviewer 1's specific comment #3, the LM4-SHARC had to be tested at a headwater catchment since the sub-grid scale reach-to-reach connectivity (i.e., channel network) is not fully operational yet in the GFDL land model, so the inflow from the upstream cannot be simulated due to absence of channel network. As addressed in lines 799-814 in the revised manuscript, directly using the streamflow/baseflow model-derived estimates available at the catchment-scale can be used to extract the NSD data at the catchment level. This could be done by assimilating observational data (in-situ, remote sensing) into hydrologic models for higher spatiotemporal runoff estimates. Lines 780-798 also argue that the local relationship between the soil properties data and recession parameters could also be used to infer groundwater diffusivity parameters downstream of the river (Hong and Mohanty, 2023b).

**References**

Hong, M., & Mohanty, B. (2023a). A new method for effective parameterization of catchment-scale aquifer through event-scale recession analysis. *Advances in Water Resources*, *174*, 104408.

Hong, M., & Mohanty, B. P. (2023b). Representing bidirectional hydraulic continuum between the stream and hillslope in the National Water Model for improved streamflow prediction. *Journal of Advances in Modeling Earth Systems*, *15*(3), e2022MS003325.

---

## Author Comment (AC2)

**Note)** Line numbers correspond to the non-tracked changes in the copy of the manuscript.

**Reviewer Comments 1 (RC1)**

The manuscript presents LM4-SHARC v1.0, a novel parameterization scheme that integrates catchment-scale soil, groundwater, and river interactions into the GFDL land model. This study addresses a critical limitation in current Earth System Models (ESMs) by improving hydrologic predictions and enhancing the representation of energy-water flux dynamics. The results, validated using observational data from Providence Creek, are compelling. I support publication with minor revisions. Below are my detailed comments:

**General Comments**

The manuscript's most significant issue lies in the methods section, which lacks clarity despite being central to the paper. From my understanding, the model uses baseflow observations and Equation (19) to derive parameters a and b, which are then used in Equations (20) and (21) to determine other parameters. During the simulation, Equation (1) calculates ql, and Equation (6) calculates Q. However, the purpose of the other equations remains unclear. The authors should explicitly outline the role of each equation, the model's calibration process, and the input-output structure. Additionally, the distinction between analytical and numerical solutions is difficult to grasp and warrants a better explanation.

- We appreciate the reviewer's helpful comments. According to the reviewer's comments, we have added a new workflow diagram that clarifies the processes to determine the catchment-scale groundwater parameters in LM4-SHARC (Figure 2). This diagram explains how the combined use of analytical/numerical models with observational data is concluded in groundwater parameterization. Also, we have revised the manuscript to clarify each equation's role, especially Equations (4), (5), (6), (7), (8), (20), and (21). Please find them in the revised manuscript.

Furthermore, the section on energy transport methods could be shortened, as the hydrological components are more critical to the study.

- We appreciate the reviewer's comments. Based on the reviewer's suggestion, we have updated section 2.3.2 to improve the readability. Specifically, explicit equations for soil, groundwater, and river temperature estimation and heat flux were retained in the main text. In contrast, all other equations (i.e., dynamic heat capacity and details of heat flux estimation) were categorized as Appendix A and excluded from the main text. As a result, section 2.3.2 has been significantly shortened while addressing the essential information of temperature simulation in LM4-SHARC.

Regarding model evaluation, while improvements are demonstrated for the Providence Creek catchment, they could stem from calibration against local streamflow data. Without such data in other regions, how can the authors ensure similar improvements? The discussion on future global tuning using remote sensing data is insufficient. Since this scheme is intended for global implementation, the manuscript should address how effective parameters can be derived at a global scale at this time. Expanding this section would significantly strengthen the manuscript.

- We appreciate the reviewer's insightful comments. Based on the reviewer's suggestions, we have extensively revised section 4.5.2 to expand the discussion on the future global tuning of the new groundwater domain in LM4-SHARC. In the revised manuscript, we addressed (1) the possibility

of using global static soil properties data such as GSDE and SoilGrids to parameterize the ungauged catchments' Boussinesq-based groundwater, (2) using machine learning-based recession parameter inference by incorporating remote-sensing and global in-situ datasets. We aim to explore if the catchment-scale streamflow recession behavior could be predicted by the remote-sensing data based on the learned relationship in-situ-derived recession characteristics and remote-sensing data. In addition to identifying the relationship between data and recession parameters, we also addressed (3) directly using the streamflow/baseflow model-derived outputs available at the catchment-scale to calculate the net subsurface discharge (NSD) for estimating the Boussinesq aquifer's diffusivity downstream of the river. Please find lines 780-798.

- All the specific comments made by the reviewer have been addressed in the following section.

**Specific Comments**

* Figure number has been changed due to the added Figure 2 – workflow diagram.

1. **L86:** "The properties of the groundwater were…"
   If soil and bedrock types are stable, why should groundwater parameters vary over time? The reasoning for developing time-varying parameters is not sufficiently convincing.

- According to recent studies, the hydraulic diffusivity of the catchment-scale groundwater is estimated differently across individual recession events (Bart & Hope, 2014; Hong & Mohanty, 2023a; Jachens et al., 2020; Li & Ameli, 2022; Trotter et al., 2024). This is mainly because the groundwater storage changes between the events (i.e., antecedent condition – transient storage) and the distinct spatial extent of the saturated groundwater zone in each recession event result in different effective properties. Groundwater storage varies across different events and can largely be attributed to climate-related factors (e.g., inter-arrival time, precipitation magnitude) and anthropogenic factors (e.g., land use change, groundwater pumping). Based on the reviewer's comments, the sentence (lines 87-89) has been updated in the revised manuscript to clarify the need to consider the groundwater parameters as emergent and dynamic properties.

2. **L111:** "Then, the catchment-scale hydrologic structure…"
   What is meant by "inter-tile connection"? Has this been fully implemented in the land model?

- Inter-tile connection means lateral divergences of fluxes between adjacent tiles as fully implemented in the land model. To improve readability, the sentence has been moved to section 2.2, and the corresponding paragraph has been updated (lines 178-180).

3. **L127:** "Sierra National Forest…"
   What criteria led to the selection of this catchment?

- First, in implementing the LM4-SHARC at the catchment-scale, it was required to select a 'headwater' catchment since the sub-grid scale reach-to-reach connectivity (i.e., channel network) is not fully operational yet in the GFDL land model, so the inflow from the upstream cannot be simulated due to absence of channel network. Moreover, the daily streamflow observations at the headwater catchment's outlet were needed to perform the recession analysis. The near-surface soil

moisture and soil temperature observations were also required to evaluate LM4-SHARC. We found that the necessary data (near surface water-energy, and streamflow) were publicly available by the Critical Zone Observatory (CZO) at the Sierra Nevada Providence Creek while successfully measuring the headwater reach's streamflow hydrograph unaffected by the upstream or any surrounding tributaries.

4. **L190-205:**
   This paragraph is unclear. The relationship between Equations (4)(5)(6) and Equations (2)(19), as well as their roles within the model, should be explicitly explained.

- According to the reviewer's comments, we have extensively updated the corresponding section (line 222-238) to clarify each equation's role and relationship in resolving streamflow dynamics at reach level.

5. **L402:** "The k was set to 1…"
   Why was k=1? Even in a headwater catchment, different types of hillslopes could exist.

- The characteristic hillslopes (CH) are defined at the catchment level. Once a specific catchment is determined to belong to a particular CH, it is further divided into multiple hillslopes depending on whether it is a headwater catchment or a downstream catchment. If the catchment is a headwater catchment, it has three hillslopes; if the catchment is downstream, it has two hillslopes. For clarification, we have added a figure regarding how the hillslopes are determined in the study catchment as Supplemental Information S.1.

6. **L541:** "Baseflow observation…"
   The baseflow observation method derived from Szilagyi and Parlange (1998) should include a discussion of its accuracy. Since baseflow is a minor yet essential component in model calibration, its uncertainty warrants consideration.

- Following the reviewer's comments, we have discussed the potential uncertainty of the baseflow estimates and separation method used in this study. In the revised manuscript, we addressed that Szilagyi method's uncertainty might be sourced from (1) the $A$ parameter (i.e., contributing area) assumed to equal to the entire catchment area and be static across recession events, and (2) the initial saturated groundwater thickness $N_{ini}$ assumed to be same across recession events at 10 m. However, due to the lack of observationally-derived $A$ and $N_{ini}$ values, this study conducted baseflow separation under these assumptions for the two parameters (lines 440-450).

7. **L576:** "The evaluation was performed for four years…"
   Why wasn't the model evaluated against streamflow/baseflow observations?

- The entire 11 years of streamflow data for the study catchment were used for calibration (WY2004-WY2014), ensuring calibration of groundwater diffusivity parameters accounts for the study area's distinct hydrometeorological conditions (e.g., dry and wet years). Figure 7 shows the enhanced baseflow estimates found in the calibration process, as explained in section 4.2.2, as well as the added workflow diagram (Figure 2). Then, the LM4-SHARC was evaluated in terms of soil moisture and temperature to show the relevance of using streamflow-derived groundwater properties to parameterize groundwater in order to predict near-surface soil water and energy

components, which is the primary focus of this study as well as climate models. Lines 587, 590-591 have been updated to this clarification.

**Conclusion**

This manuscript is a valuable contribution to Earth system modeling. With improved clarity in the methods section, expanded discussion on global parameterization, and more detailed explanations for specific concerns, it would be well-suited for publication.

- We truly appreciate the reviewer's encouraging comments.

**References**

Bart, R., & Hope, A. (2014). Inter-seasonal variability in baseflow recession rates: The role of aquifer antecedent storage in central California watersheds. *Journal of Hydrology*, *519*, 205-213.

Hong, M., & Mohanty, B. (2023a). A new method for effective parameterization of catchment-scale aquifer through event-scale recession analysis. *Advances in Water Resources*, *174*, 104408.

Jachens, E. R., Rupp, D. E., Roques, C., & Selker, J. S. (2020). Recession analysis revisited: impacts of climate on parameter estimation. *Hydrology and Earth System Sciences*, *24*(3), 1159-1170. https://doi.org/10.5194/hess-24-1159-2020

Li, H., & Ameli, A. (2022). A statistical approach for identifying factors governing streamflow recession behaviour. *Hydrological Processes*, *36*(10), e14718.

Trotter, L., Saft, M., Peel, M. C., & Fowler, K. J. (2024). Recession constants are non-stationary: Impacts of multi-annual drought on catchment recession behaviour and storage dynamics. *Journal of Hydrology*, *630*, 130707.